# BOUNDARY-AWARE TOKENIZATION FOR EVENT-DRIVEN TIME-SERIES FORECASTING

## ABSTRACT

Transformer-based large sequence models have recently been extended from language to time-series to capture long-range dependencies and heterogeneous dynamics. However, unlike language, time-series lack a natural dictionary for principled tokenization: existing large sequence models often resort to fixed-length tokens or patches for computational efficiency. This design can obscure regime changes, expend attention on low-information tokens, and restrict the effective context length. We address this limitation with Boundary-aware tokenization, which initiates new tokens only at predicted regime changes in the time-series, analogous to how spaces delimit words in language. At its core, the model integrates an unsupervised boundary detector to form variable-length chunks, an intra-chunk fusion module to derive chunk-level token embeddings, and a smoothing module to stabilize training, before passing the resulting tokens to Transformer-based modules. We further add a gating refinement that fuses fixed- and variable-length representations before the forecasting decoder, enabling adaptive selection during pre-training based on data patterns. This design directly addresses event-driven regime changes, while remaining robust in stationary regimes. Across diverse benchmarks, our method reduces forecasting error by 10.5% on average, with learned chunks aligned with true regime boundaries. We also show that the model adaptively reverts to fixed-length tokenization in stationary time-series.

## 1 INTRODUCTIONN

Many real-world time-series are not deterministic ODE systems, where trajectories can be fully recovered from initial conditions. Instead, they are shaped by external events, e.g., stock prices reacting to market shocks (Long et al., 2024) or electricity demand spiking under extreme weather (Xiao et al., 2023). Such event-driven patterns create high-impact regime changes that remain challenging for traditional forecasters (Oliveira & Ramos, 2024). While recent efforts adapt Transformer-based large sequence models to capture long-range dependencies and heterogeneous dynamics (e.g., Informer (Zhou et al., 2021), FEDformer (Zhou et al., 2022), PatchTST (Nie et al., 2022)), these strong models still allocate tokens *uniformly*: either every step becomes a token or fixed-length patches are used (Ansari et al., 2024; Garza et al., 2023; Jin et al., 2023). Such fixed-length tokenization allocates equal capacity to quiescent spans and sharp transitions, misaligning representation with events: in solar ramps, ICU alarms, or EV fast-charging spikes, rare but critical transitions are underemphasized, while redundant intervals dominate the token budget.

Across modalities, tokenization choices strongly bias downstream behavior (Hwang et al., 2025; Singh & Strouse, 2024). We argue that time-series deserve the same dictionary privilege that benefited language, such as the carefully engineered BPE vocabulary (Sennrich et al., 2015). Unlike language, where a finite dictionary can capture most patterns, time-series lack such a predefined vocabulary. This calls for a learned front-end that adaptively groups time steps into tokens where information concentrates, rather than relying on fixed-length tokenization chosen merely for computational efficiency. This motivates our boundary-aware Boundary-aware tokenization, which creates new tokens only at predicted regime changes in the time-series, analogous to how spaces delimit words in language.

To realize this idea, our architecture integrates three key modules. (i) Boundary detector: an unsupervised predictor that leverages local embedding dynamics to identify regime changes, segmenting

the sequence into variable-length chunks aligned with event boundaries. This design is inspired by recent advances in dynamic tokenization for natural language (Hwang et al., 2025), but is adapted to the unique structure of time-series. (ii) Chunk-level embedding: within each chunk, a mixture-of-experts (Masoudnia & Ebrahimpour, 2014) fusion combines complementary statistics (mean, boundary, min/max, and attention pooling) into a compact token representation enriched with positional metadata. (iii) Chunk smoothing: a causal exponential moving average refines chunk embeddings using boundary confidences, blending uncertain transitions while preserving sharp regime shifts. These event-aware tokens are then processed by causal Transformers (Vaswani et al., 2017) to capture long-range dependencies. To restore predictions to the original resolution, a cross-attention decoder aligns future time queries with past chunk representations. The resulting *Boundary-aware Tokenization Large Signal Model* (BT-LSM) that concentrates capacity where dynamics change, preserves accuracy at evaluation, and decodes from a compact event-level sequence.

Not all data patterns benefit from variable-length tokenization. To address this, we introduce a gating refinement that safely combines fixed-length and variable-length strategies. Both representations are retained, and a lightweight gate adaptively selects or fuses them during training based on data patterns. This design ensures robustness: in stationary or coarse-grained regimes, the model naturally reverts to fixed-length behavior (as in PatchTST Nie et al. (2022)), while in bursty or irregular regimes, the gate activates variable-length adjustments to capture critical transitions (see Section 4.4 for empirical validation).

Our contributions are summarized as follows:

- We propose Boundary-aware Tokenization Large Signal Model (BT-LSM) for time-series, introducing a lightweight unsupervised boundary detector and mixture-of-experts chunk embeddings. Our boundary-aware model allocates tokens adaptively to event-driven transitions, avoiding uniform waste. We also show a resampling-invariance theorem of such tokenization process.

- We design a gating refinement that adaptively fuses fixed-length and variable-length representations, ensuring robustness across data regimes and yielding performance comparable to strong fixed-length baselines in stationary settings. In Section 4.4, we show that BT-LSM adaptively reverts to fixed-length tokenization in stationary time-series.

- We demonstrate that BT-LSM achieves over 10.5% lower forecasting error at matched compute budgets across diverse benchmarks, including energy, power, and traffic data. It addresses bursty, event-driven regime changes where uniform tokenization fails.

## 2 RELATED WORK

**Transformer-based models for time-series.** Transformer architectures have recently emerged as the dominant backbone for long-horizon time-series forecasting. Informer (Zhou et al., 2021) introduced ProbSparse attention to scale self-attention to longer sequences. FEDformer (Zhou et al., 2022) combined frequency-domain decomposition with Transformer blocks to improve efficiency and robustness. PatchTST (Nie et al., 2022) borrowed ideas from vision Transformers by segmenting time-series into fixed-size patches treated as tokens. Beyond forecasting, models such as Autoformer (Wu et al., 2021), Reformer (Kitaev et al., 2020), and TimesNet (Wu et al., 2022) further extended Transformer-based design for multi-scale temporal patterns. Despite their architectural diversity, they inherit NLP-style fixed tokenization for computation efficiency. Our work is the first to address event-driven tokenization with native-grid fidelity under resampling-invariance guarantees.

**Beyond fixed-length tokenization.** Several approaches have attempted to move beyond uniform tokenization. Some rely on signal decomposition or segmentation: for example, SIMTSeg (Bao et al., 2024) and U-Time (Perslev et al., 2019) learn boundaries via supervised segmentation, but require external labels or domain-specific priors. Other methods employ adaptive discretization of time or values, such as neural segmentation models in speech (Wang et al., 2017; Chung et al., 2016), or dynamic tokenization in NLP (Hwang et al., 2025). However, these designs are either tied to supervised tasks, or inherit discretization heuristics rather than addressing forecasting directly. Closest to our work are efforts treating time-series as "language" (Ansari et al., 2024; Garza et al., 2023; Jin et al., 2023), yet they typically adopt fixed-length or handcrafted tokenization borrowed from NLP pipelines. In contrast, our approach introduces a Boundary-aware tokenization tailored to forecasting: an unsupervised boundary detector aligned with local dynamics.

## 3 METHOD

**Problem setup.** We study multivariate time series forecasting, a fundamental task in machine learning with wide applications in domains such as energy (Heidrich et al., 2020; Lara-Benítez et al., 2020), finance (Wu et al., 2020; Zeng et al., 2023), and healthcare (Morid et al., 2023; Song et al., 2024). Let $\mathbf{x}_{1:T} = [\mathbf{x}_1, \mathbf{x}_2, \ldots, \mathbf{x}_T]^\top \in \mathbb{R}^{T \times D}$ denote an input sequence of length $T$, where each observation $\mathbf{x}_t \in \mathbb{R}^D$ is a $D$-dimensional vector. The goal is to forecast the next $H$ future steps. Throughout, we assume a causal setting: the predictor at time $t$ may only depend on the past observations $\mathbf{x}_{1:t-1}$.

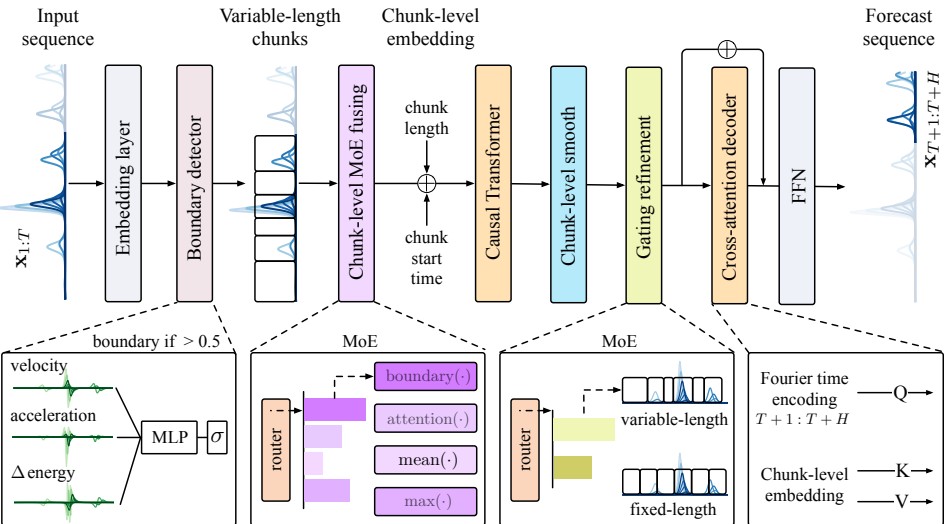

Figure 1: Model architecture of BT-LSM.

As illustrated in Figure. 1, our framework consists of five stages. (i) An *embedding and boundary detection stage* projects raw observations $\mathbf{x}_{1:T}$ into latent vectors $\mathbf{e}_{1:T}$ and predicts event boundaries using a boundary detector, which segments the sequence into variable-length chunks. (ii) A *chunk-level embedding stage* aggregates within-chunk embeddings into chunk tokens $\mathbf{z}_m$ through a mixture-of-experts fusing strategy (boundary, attention, mean, and max-norm pooling), augmented with chunk length and position metadata. (iii) A *chunk-level sequence modeling stage* processes the resulting chunk tokens with a stack of causal Transformer decoder blocks, followed by a causal EMA-based smoothing module that stabilizes transitions across uncertain boundaries and ensures differentiability. (iv) A *gating refinement stage* fuses the variable-length representations with a parallel fixed-length tokenization stream, enabling adaptive selection between the two to yield performance comparable to strong fixed-length baselines in stationary settings. (v) Finally, a *cross-attention decoding stage* uses Fourier embeddings of future time steps as queries, with the smoothed chunk tokens as keys and values, to generate the final step-level forecasts $\hat{\mathbf{x}}_{T+1:T+H}$.

### 3.1 BT-LSM ARCHITECTURE

**Embedding layer for enhanced representational capacity.** We first normalize all training sequence data $\mathbf{x}_{1:T+H}$ to zero mean and unit variance to stabilize optimization and make features comparable across dimensions. For input sequence $\mathbf{x}_{1:T}$, each observation $\mathbf{x}_t \in \mathbb{R}^D$ is projected into a high-dimensional latent representation: $\mathbf{e}_t = \mathbf{W}_x \mathbf{x}_t \in \mathbb{R}^{D_e}$, $t = 1, \ldots, T$, where $\mathbf{W}_x \in \mathbb{R}^{D_e \times D}$ is a learnable linear projection. This embedding layer serves two purposes: (i) it unifies heterogeneous input features into a shared latent space, and (ii) it expands the representational capacity ($D_e \gg D$), enabling downstream modules to capture higher-order temporal dependencies more effectively (Vaswani et al., 2017; Bai et al., 2018).

**Boundary detector for event-driven segmentation.** A key challenge in time-series modeling is that many real-world sequences exhibit *event-based patterns*, where natural segment boundaries carry semantic meaning, such as in physiological signals or regime shifts in sensor and financial data (Perslev et al., 2019; Monteiro & Costa, 2023). Unlike natural language, where a pre-defined

dictionary of tokens provides a well-established segmentation, fixed-length tokenization in time series may inadvertently merge heterogeneous regimes into a single token. For example, a fixed-length window that straddles an event transition may fuse pre- and post-event dynamics, thereby blurring meaningful information. To overcome this limitation, we introduce a boundary prediction module that adaptively infers token boundaries aligned with the intrinsic structure of the sequence. Our design is inspired by recent advances in dynamic tokenization for natural language (Hwang et al., 2025), but is tailored to the time-series domain. In contrast to prior neural segmentation approaches such as U-Time (Perslev et al., 2019) and SIMTSeg (Bao et al., 2024), which classify each time point into segments or rely on downstream supervision, our method explicitly leverages first- and second-order variations of learned embeddings as local signals for boundary detection.

Concretely, given the embedding sequence $\mathbf{e}_{1:T}$, we compute three temporal signals to capture local dynamics: the velocity $\mathbf{v}_t = \mathbf{e}_{t+1} - \mathbf{e}_t \in \mathbb{R}^{D_e}$, the acceleration $\mathbf{a}_t = \mathbf{v}_{t+1} - \mathbf{v}_t \in \mathbb{R}^{D_e}$, and the energy change $\Delta \mathcal{E}_t = \|\mathbf{e}_{t+1}\|_2 - \|\mathbf{e}_t\|_2 \in \mathbb{R}$. Each quantity is projected into a latent space via learnable matrices $\mathbf{W}_v, \mathbf{W}_a, \mathbf{W}_{\mathcal{E}}$, combined additively, and passed through a sigmoid activation to yield a boundary probability as in Eq. (1). Here, $p_t$ quantifies the likelihood that position $t$ corresponds to a meaningful boundary. Following Hwang et al. (2025), we obtain a hard boundary decision via $b_t = \mathbb{1}\{p_t \geq 0.5\}$, where $\mathbb{1}\{\cdot\}$ denotes the indicator function. We additionally enforce $b_1 = 1$ to guarantee a boundary at the beginning of the sequence.

$$p_t = \text{Sigmoid}(\mathbf{W}_v \mathbf{v}_t + \mathbf{W}_a \mathbf{a}_t + \mathbf{W}_{\mathcal{E}} \Delta \mathcal{E}_t), \qquad p_t \in (0, 1). \qquad (1)$$

A technical challenge arises because $b_t$ is obtained through a non-differentiable thresholding operation, for which $\partial b_t / \partial p_t = 0$ almost everywhere. To preserve trainability, we adopt a straight-through estimator (STE) (Bengio et al., 2013): during the forward pass we use the discrete $b_t$, while during the backward pass we approximate the gradient as $\partial b_t / \partial p_t \approx 1$. Consequently, the loss gradient with respect to $p_t$ reduces to $\frac{\partial \mathcal{L}}{\partial p_t} = \frac{\partial \mathcal{L}}{\partial b_t}$, allowing gradients to flow continuously through $p_t$ during training while retaining discrete boundary decisions at inference.

**Chunk-level embedding based on Mix-of-Experts fusing.** Once boundaries are predicted, we segment the sequence into contiguous *chunks*, where each chunk corresponds to the subsequence between two boundaries. Let $M$ denote the total number of chunks. Specifically, chunk $m$ spans from $t^{(m)}$ to $t^{(m+1)}$, with $t^{(m)}$ denoting the start index of chunk $m$ and $t^{(M+1)} = T$ ensuring full coverage of the sequence. Unlike fixed-length tokens, these variable-length chunks preserve event-level semantics in time-series data.

Each chunk $m$ contains a variable number of time steps, represented by embeddings $\mathbf{e}_{t^{(m)}:t^{(m+1)}}$. To obtain a single chunk-level embedding $\mathbf{z}_m$, we employ a mixture-of-experts (MoE) strategy (Masoudnia & Ebrahimpour, 2014) that combines four complementary fusion mechanisms. (1) The boundary embedding is defined as $\mathbf{z}_m^{\text{boundary}} = \mathbf{e}_{t^{(m)}}$, preserving the representation at the regime transition. (2) The attention pooling is given by $\mathbf{z}_m^{\text{attn}} = \sum_{t \in C_m} \alpha_t \mathbf{e}_t$, where $\alpha_t = \text{softmax}(\mathbf{w}^\top \mathbf{e}_t)$, adaptively weighting embeddings by their learned relevance. (3) The mean pooling computes $\mathbf{z}_m^{\text{mean}} = \frac{1}{t^{(m+1)} - t^{(m)}} \sum_{t=t^{(m)}}^{t^{(m+1)}} \mathbf{e}_t$, capturing the average trend within the chunk. (4) The max pooling is defined as $\mathbf{z}_m^{\text{max}} = \arg\max_{\mathbf{e}_t, t \in [t^{(m)}, t^{(m+1)}]} \|\mathbf{e}_t\|_2$, selecting the most dominant embedding to highlight the most informative time step. These expert outputs are combined via a learnable gating mechanism in Eq. (2), where $\{g_k\}$ are softmax-normalized mixture weights and $\{\gamma_k\}$ are trainable parameters. This MoE formulation adaptively balances local statistics, boundary sensitivity, and attention-driven fusion within each chunk.

$$\mathbf{z}_m^{\text{MoE}} = \sum_{k \in \{\text{mean,boundary,max,attn}\}} g_k \, \mathbf{z}_m^k, \qquad g_k = \frac{\exp(\gamma_k)}{\sum_j \exp(\gamma_j)}, \qquad (2)$$

Finally, to encode structural information, we concatenate two metadata to each fused embedding: the chunk length $(t^{(m+1)} - t^{(m)})$ and the start position $t^{(m)}$. These provide the model with explicit knowledge of *where* the chunk occurs in the sequence and *how long* it spans. The resulting chunk representation has dimensionality $D_z = D_e + 2$: $\mathbf{z}_m = \left[ \mathbf{z}_m^{\text{MoE}} \parallel (t^{(m+1)} - t^{(m)}) \parallel t^{(m)} \right] \in \mathbb{R}^{D_z}$.

**Chunk-level causal Transformer for long-range dependency modeling.** To capture long-range dependencies across chunks, we process the sequence of chunk tokens $\mathbf{z}_{1:M}$ with a stack of $L$

*causal Transformer decoder* blocks. The initial hidden state for each chunk is set directly from its embedding: $\mathbf{h}_m^0 = \mathbf{z}_m$. For $\ell = 1, \ldots, L$, we apply pre-norm residual updates:

$$\tilde{\mathbf{h}}_m^\ell = \mathbf{h}_m^{\ell-1} + \text{MHA}_\ell\big(\text{LN}(\mathbf{h}_{\leq m}^{\ell-1})\big), \quad \mathbf{h}_m^\ell = \tilde{\mathbf{h}}_m^\ell + \text{FFN}_{\tanh,\ell}\big(\text{LN}(\tilde{\mathbf{h}}_m^\ell)\big), \tag{3}$$

where $\text{MHA}_\ell$ is masked (causal) multi-head self-attention, LN denotes LayerNorm, and $\text{FFN}_{\tanh,\ell}$ is a position-wise feed-forward network with a $\tanh$ nonlinearity. The top-layer state $\mathbf{h}_m^L$ is mapped to a prediction through a lightweight feed-forward projection: $\hat{\mathbf{z}}_m = \text{FFN}_{\text{out}}\big(\mathbf{h}_m^L\big) \in \mathbb{R}^{D_z}$.

**Chunk smoothing for stable boundary transitions.** In the boundary prediction module, discrete chunk decisions were obtained from the hard thresholded variables $b_m$. To better leverage the underlying uncertainty, we refine chunk representations using the soft boundary likelihoods $p_m$. Specifically, let $\{\tilde{p}_m\}_{m=1}^M$ denote the downsampled boundary confidences (with $\tilde{p}_m = p_{s_m}$ at predicted start indices $s_m$). We then apply a causal exponential moving average (EMA) (Hwang et al., 2025) to produce boundary-aware latent states:

$$\bar{\mathbf{z}}_m = (1 - \tilde{p}_m)\bar{\mathbf{z}}_{m-1} + \tilde{p}_m \hat{\mathbf{z}}_m, \quad m = 2, \ldots, M, \quad \bar{\mathbf{z}}_1 = \hat{\mathbf{z}}_1. \tag{4}$$

Intuitively, high-confidence boundaries ($\tilde{p}_m \approx 1$) allow the new chunk embedding $\hat{\mathbf{z}}_m$ to pass through with little mixing, while uncertain boundaries ($\tilde{p}_m \approx 0$) blend more smoothly with the preceding state $\bar{\mathbf{z}}_{m-1}$. This parameter-free, strictly causal smoothing mechanism stabilizes transitions across chunk boundaries during training (See Sec. 4.3 for an ablation study comparing models with and without chunk smoothing).

**Gating refinement for fixed- and variable-length fusion.** Not all time-series benefit equally from variable-length tokenization, which is primarily designed for event-driven dynamics. To ensure robustness, we introduce a gating module that fuses both representations: a *variable-length* representation $\bar{\mathbf{z}}_m^{\text{var}}$ from Eq. (4), and a *fixed-length* representation $\bar{\mathbf{z}}_m^{\text{fix}}$ obtained through the same pipeline under a fixed-length tokenization strategy. The final chunk representation is produced by a mixture-of-experts gate: $\bar{\mathbf{z}}_m = \sum_{k \in \{\text{var,fix}\}} g_k \bar{\mathbf{z}}_m^k$, where $\{g_k\}$ are softmax-normalized mixture weights and $\{\gamma_k\}$ are trainable parameters. This formulation allows the model to adaptively interpolate between fixed- and variable-length representations on a per-chunk basis, preserving the strong baseline behavior of fixed-length tokenization in stationary regimes while exploiting boundary-aware representations in bursty or irregular regimes.

**Cross-attention decoder for step-level forecasting.** We forecast future observations by aligning chunk-level representations with target time queries. We first encode the horizon of interest $T+1 : T+H$ using Fourier time embeddings (Vaswani et al., 2017): $\mathbf{q}_\tau = \text{FourierEmbed}(\tau), \tau = T+1, \ldots, T+H$, which serve as the *queries* in a cross-attention layer. These embeddings provide a continuous representation of future time points, enabling the model to generalize across arbitrary prediction horizons. The smoothed chunk-level embeddings $\{\bar{\mathbf{z}}_m\}_{m=1}^M$ are used as both *keys* and *values*. Formally, the cross-attention decoder computes $\mathbf{h}_\tau = \text{MHA}\big(\mathbf{q}_\tau, \{\bar{\mathbf{z}}_m\}_{m=1}^M, \{\bar{\mathbf{z}}_m\}_{m=1}^M\big)$, where $\text{MHA}(\cdot)$ denotes multi-head attention (MHA). Intuitively, each future time step $\tau$ attends to past chunks in proportion to their learned temporal relevance. Finally, the attention output is mapped through a lightweight feed-forward layer to produce the predicted observation: $\hat{\mathbf{x}}_\tau = \text{FFN}_{\text{out}}(\mathbf{h}_\tau), \tau = T+1, \ldots, T+H$. This design allows the decoder to directly condition predictions on temporally localized, event-aware chunk representations while maintaining flexibility to extrapolate across variable-length horizons.

**Batch computation via padding.** While variable-length chunks provide stronger representational power for capturing event-based structure in time-series data, they introduce a practical challenge: unlike fixed-length tokens, variable-length chunks cannot be directly processed in parallel batches. To address this, we adopt a padding-and-masking strategy. Specifically, we pad the number of chunks in each sequence to a maximum $M^{\max}$, determined by the largest number of chunks in the batch, and pad the time steps within each chunk to a maximum length $l^{\max}$. Zero values are used for padding, and binary masks indicate positions that correspond to padded (non-informative) entries. This enables the model to recover full batch parallelism while preserving the semantics of variable-length chunking.

**Training loss.** The training objective combines forecasting accuracy with boundary and continuity regularization: $\mathcal{L} = \mathcal{L}_{\text{pred}} + \lambda_{\text{boundary}} \mathcal{L}_{\text{boundary}} + \lambda_{\text{cont}} \mathcal{L}_{\text{cont}}$, where $\mathcal{L}_{\text{pred}} = \frac{1}{H} \sum_{h=1}^{H} \|\mathbf{x}_{T+h} - \hat{\mathbf{x}}_{T+h}\|_2^2$, $\mathcal{L}_{\text{boundary}} = \left( \frac{1}{T} \sum_{t=1}^{T} b_t - \frac{1}{\kappa} \right)^2$, and $\mathcal{L}_{\text{cont}} = \|\hat{\mathbf{x}}_{T+1} - \mathbf{x}_T\|_2^2$. Here $\mathcal{L}_{\text{pred}}$ enforces accurate multi-step forecasting, $\mathcal{L}_{\text{boundary}}$ constrains the expected boundary rate to the target compression $\kappa$, and $\mathcal{L}_{\text{cont}}$ discourages discontinuities at the forecast interface. Appendix C details the full training procedure.

**Invariance to intra-chunk resampling.** In many real-world time-series, sampling rates are heterogeneous (e.g., sensor-dependent) or irregular (e.g., due to missing data). If the chunk-level embedding were sensitive to the number of points observed within a chunk, then simple re-sampling or interpolation could distort the learned dynamics and destabilize forecasting. Hence, it is desirable that once chunk boundaries are fixed, the resulting chunk-level embedding and the model's forecasts, remain unchanged under any re-sampling inside the chunk. Formally, such a re-sampling can be expressed as a monotone time warp that stretches or compresses the points within a chunk while preserving its endpoints. In Theorem 1, we establish that BT-LSM satisfies this invariance, with proof in Appendix D.

**Theorem 1** (Invariance to intra-chunk resampling). *Let $\phi_m : [t^{(m)}, t^{(m+1)}] \to [t^{(m)}, t^{(m+1)}]$ be any monotone time warp with fixed endpoints, and define the warped embeddings $\tilde{\mathbf{e}}(t) := \mathbf{e}(\phi_m(t))$ for $t \in [t^{(m)}, t^{(m+1)}]$. Then, under Assumption 1 (content-only experts) and Assumption 2 (bounded embeddings), the resulting chunk representation is unchanged: $\tilde{\mathbf{z}}_m = \mathbf{z}_m$, and consequently the model's forecasts remain identical: $\tilde{\hat{\mathbf{x}}}_{T+1:T+H} = \hat{\mathbf{x}}_{T+1:T+H}$.*

# 4 Numerical Results

**Dataset.** We evaluate our model and other cutting-edge methods on seven diverse datasets: (1) *Building* Addison et al. (2019) with ASHRAE building energy and temperature data, (2) *Spain* Kolasniwash (2019) with four years of national electricity and weather data, (3) *Consumption* Fedesoriano (2022) with 52,416 ten-minute records from Tetouan city including meteorology features, (4) *Residential* Sri Polu (2019) with hourly household usage and weather in Houston, (5) *Solar* AI Maverick (2023) with renewable energy and temperature data, (6) *ETT* Zhou et al. (2021) with hourly transformer temperature series (*ETTh1*, *ETTh2*), and (7) *Traffic* Lai et al. (2018) with freeway occupancy data. For all datasets, we split sequences by week and train models to forecast the next 24 hours from historical inputs, with additional horizons (12/48) reported in Appendix F.1.

**Experimental setup.** All experiments were conducted on an HPC cluster with NVIDIA A100 GPUs (80 GB), using Python 3.9 and PyTorch implementations of all baselines (Informer Zhou et al. (2021), Contiformer Chen et al. (2023), ODE-RNN (Rubanova et al., 2019) RNN-$\Delta t$ (Che et al., 2018) PatchTST (Nie et al., 2022) Chronos (Ansari et al., 2024) GPT (Radford et al., 2019)) and our proposed method. Each baseline was configured following its canonical design. Our model BT-LSM was instantiated with width 128, six self-attention layers, eight heads, and a maximum of 50 chunks. Optimization used AdamW with cosine annealing and gradient clipping. Further implementation details, hardware specifications, and training schedules are provided in Appendix B.

## 4.1 Demonstration of Variable-length Chunking

BT-LSM converts dense sequences into compact, event-aligned token streams. As shown in Figure 2, boundaries in univariate series concentrate near spikes and regime shifts, while in multivariate traffic they synchronize across dimensions to isolate shared peaks. Appendix E further shows that in *Spanish/Building Energy & Temp*, boundaries align with cycle inflections and forecast onsets; in bursty *Solar* and *Consumption*, chunks contract around spikes and expand in quiescent spans; and in *Residential Temp*, plateaus compress into stable chunks with a single step-change break. These behaviors confirm that the boundary detector implicitly acts as an event detector, synchronizing tokens with ramps, spikes, and alarms, which underpins the performance gains in Table 1.

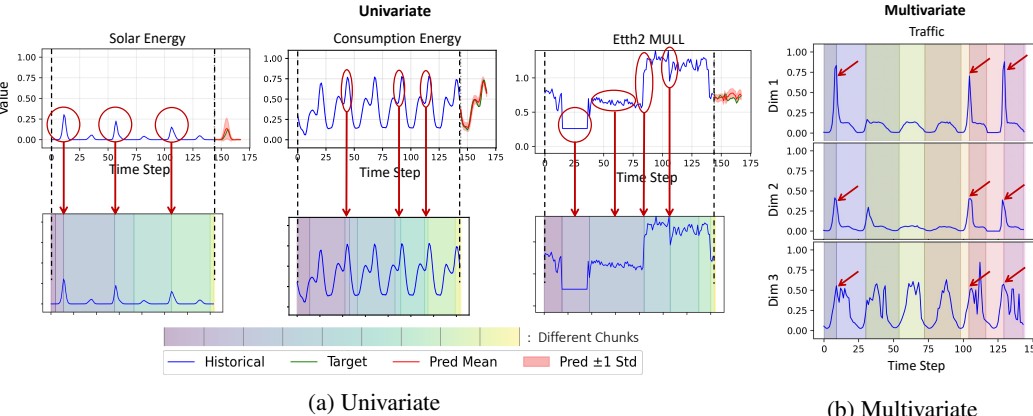

(a) Univariate

(b) Multivariate

Figure 2: In (a) univariate time-seires and (b) multivariate traffic data, the boundary detector in BT-LSM aligns tokens with spikes, inflection points, or regime changes, showing that the model learns event-synchronous tokenization.

Table 1: Performance comparison across datasets (all values in $\times 10^{-3}$).

| Method | #Para | Load | | Energy | | | Temperature | | | | | Traffic |
|---|---|---|---|---|---|---|---|---|---|---|---|---|
| | | Spanish | Residential | Solar | Building | Consumption | Spanish | Solar | Building | Consumption | Residential | |
| RNN-$\Delta t$ | 0.33M | $15.92 \pm 2.23$ | $13.57 \pm 3.03$ | $19.53 \pm 4.17$ | $8.93 \pm 1.27$ | $5.42 \pm 0.78$ | $10.94 \pm 1.55$ | $6.43 \pm 0.91$ | $17.83 \pm 2.47$ | $9.84 \pm 1.39$ | $19.86 \pm 5.59$ | $19.78 \pm 1.12$ |
| ODE-RNN | 0.53M | $18.89 \pm 2.66$ | $12.93 \pm 3.23$ | $13.07 \pm 4.36$ | $9.53 \pm 1.31$ | $5.83 \pm 0.83$ | $11.63 \pm 1.64$ | $6.83 \pm 0.96$ | $19.04 \pm 2.68$ | $10.64 \pm 1.49$ | $12.04 \pm 5.89$ | $17.40 \pm 0.40$ |
| Informer | 2.05M | $16.27 \pm 1.97$ | $10.07 \pm 2.43$ | $18.04 \pm 3.39$ | $8.23 \pm 0.99$ | $5.03 \pm 0.61$ | $10.43 \pm 1.26$ | $6.13 \pm 0.75$ | $16.53 \pm 1.97$ | $9.23 \pm 1.12$ | $18.07 \pm 4.57$ | $13.23 \pm 0.32$ |
| ContiFormer | 1.85M | $7.13 \pm 0.87$ | $12.54 \pm 1.52$ | $18.23 \pm 2.19$ | $4.23 \pm 0.51$ | $2.63 \pm 0.32$ | $5.93 \pm 0.72$ | $3.43 \pm 0.42$ | $10.13 \pm 1.23$ | $5.23 \pm 0.63$ | $14.49 \pm 2.83$ | $14.23 \pm 0.32$ |
| GPT | 19.05M | $9.13 \pm 1.08$ | $14.18 \pm 1.68$ | $15.04 \pm 2.54$ | $5.13 \pm 0.62$ | $3.13 \pm 0.38$ | $6.83 \pm 0.83$ | $3.93 \pm 0.48$ | $11.53 \pm 1.39$ | $6.13 \pm 0.74$ | $16.13 \pm 3.14$ | $11.23 \pm 0.42$ |
| Chronos | 46.15M | $14.63 \pm 1.76$ | $9.63 \pm 2.25$ | $14.03 \pm 3.14$ | $3.66 \pm 0.92$ | $2.63 \pm 0.56$ | $4.53 \pm 0.35$ | $5.63 \pm 0.68$ | $7.53 \pm 0.87$ | $4.63 \pm 1.05$ | $12.28 \pm 4.24$ | $10.12 \pm 0.31$ |
| PatchTST | 0.61M | $7.57 \pm 0.92$ | $13.23 \pm 1.57$ | $13.53 \pm 1.35$ | $4.53 \pm 0.55$ | $2.83 \pm 0.35$ | $6.23 \pm 0.75$ | $3.63 \pm 0.44$ | $10.73 \pm 1.29$ | $5.53 \pm 0.67$ | $12.63 \pm 1.97$ | $10.99 \pm 0.91$ |
| **Ours** | 0.90M | $3.97 \pm 0.24$ | $7.73 \pm 0.47$ | $12.83 \pm 0.78$ | $2.64 \pm 0.16$ | $1.61 \pm 0.10$ | $3.74 \pm 0.22$ | $2.15 \pm 0.13$ | $6.27 \pm 0.38$ | $3.23 \pm 0.19$ | $11.87 \pm 0.89$ | $9.12 \pm 0.31$ |

## 4.2 EVENT-BASED TIME SERIES FORECASTING

Many real-world signals are governed not by smooth periodicity but by abrupt event-triggered changes, such as demand surges, equipment failures, or shifts in external conditions. As shown in Figure 3, the Spanish energy data exhibits sudden jumps and plateaus that standard forecasters often overshoot or smooth out. BT-LSM addresses this challenge by leveraging dynamic chunking: boundaries are placed at regime changes, allowing forecasts to condition on tokens that explicitly encode the transition. This produces trajectories that remain close to the target path, with smoother dynamics and tighter uncertainty bands compared to step-by-step autoregressive models like GPT, which tend to drift and inflate variance. While strong baselines such as Chronos remain competitive in overall accuracy, BT-LSM consistently yields more stable predictions around event boundaries, demonstrating the advantage of aligning tokenization with event-driven structure.

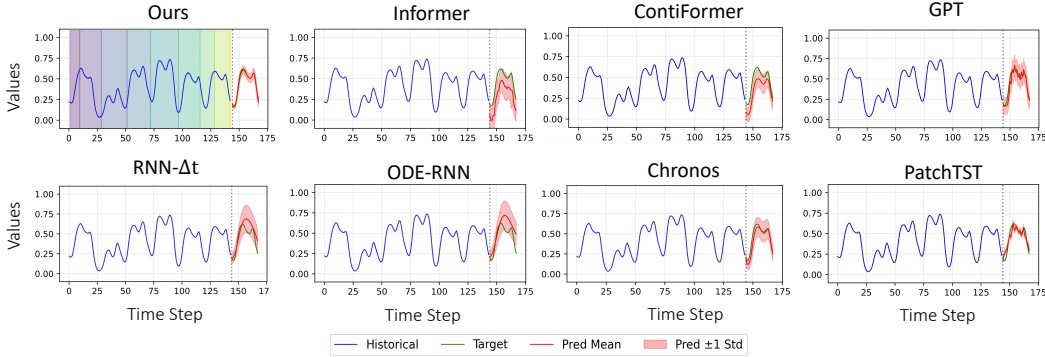

Figure 3: Comparison of forecasting in Spanish Energy dataset.

Furthermore, Figure. 4 illustrates similar challenges in ETT dataset (we show the MULL sequence as an example). ETT dataset exhibits sudden jumps and plateaus that standard forecasters struggle to capture. Our model handles such regimes more effectively by relying on dynamic chunking. When an event occurs, the boundary detector places a new chunk at the transition, allowing the subsequent

forecast to condition on a token that explicitly encodes the change. This yields forecasts that remain close to the target trajectory even after sharp jumps, with narrow uncertainty bands around event transitions. Compared to baselines, the predictions are less biased and avoid drift, demonstrating the benefit of aligning tokens with event boundaries rather than fixed-length segments.

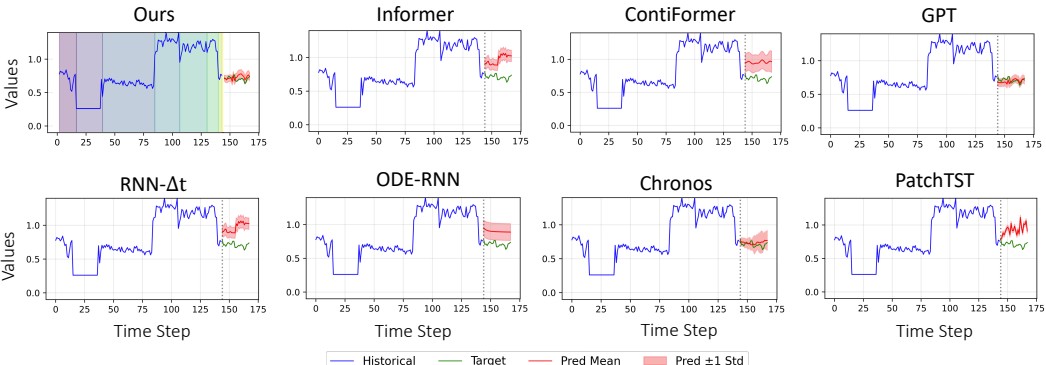

Figure 4: Comparison of forecasting in ETTh2 MULL data.

Although strong models like Chronos still show competitive accuracy, our approach better reflects the event-driven nature of the data, delivering forecasts that are both adaptive to sudden regime shifts and stable across horizons. This event-aware capability is a core advantage of boundary-based tokenization over fixed-step or patch-based alternatives. Full performance in ETT dataset accross baselines is shown in Figure 5.

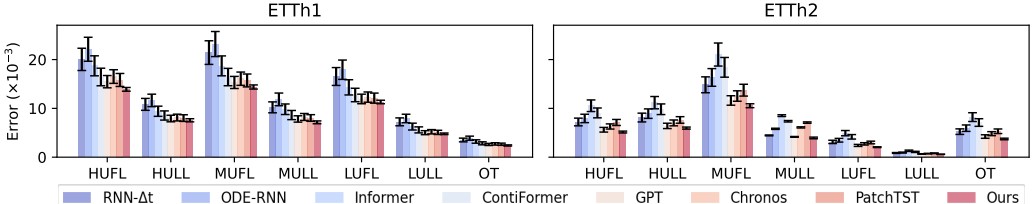

Figure 5: Forecast error on the ETT benchmark (values scaled by $\times 10^{-3}$). Left: ETTh1; right: ETTh2. Each group shows the seven targets (HUFL, HULL, MUFL, MULL, LUFL, LULL, OT).

### 4.3 ABLATION STUDY

**Ablation of model components.** Figure 6 shows forecasting error when modules are removed. We observe that dropping the boundary detector and reverting to fixed-length tokens sharply increases error. It confirms that our gains stem from detecting regime changes, not just extra parameters. Removing MoE fusion degrades accuracy by discarding complementary within-chunk statistics, while disabling EMA smoothing destabilizes boundaries. Finally, eliminating the gating refinement reduces robustness in stationary regimes, though with a smaller impact. This gating refinement ensures the event-driven path is used only when needed, yielding a safety property.

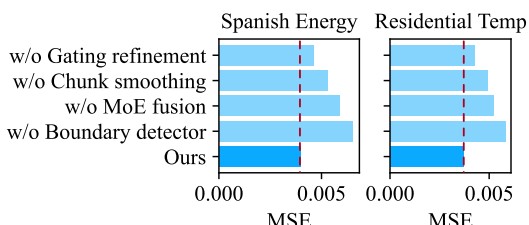

Figure 6: Results in ablation studies.

**Fixed-Length vs. Dynamic Chunking.** To further isolate the effect of variable-length tokenization, we keep the architecture and training schedule unchanged (MoE fusion, EMA smoothing, decoder, optimizer, seeds) and replace the boundary detector with fixed non-overlapping windows of length $M \in \{5, 10, 20\}$.

The token budget is matched to the dynamic model by capping to the same $M_{max}$ and masking surplus tokens; no other hyperparameters are altered. Table 2 reports forecasting MSE ($\times 10^{-3}$) on two representative datasets. Very short windows ($M = 5$) underfit longer dependencies, while long windows ($M = 20$) blend across regime changes; $M = 10$ is the strongest fixed baseline but still lags behind dynamic chunking, which compresses stationary spans and allocates more tokens around events.

Table 2: Fixed-length vs. dynamic chunking (MSE $\times 10^{-3}$).

| Method | Spanish Energy | Residential Temp |
|---|---|---|
| Fixed-5 | 5.48 | 4.98 |
| Fixed-10 | 4.58 | 4.31 |
| Fixed-20 | 4.97 | 4.74 |
| Dynamic | **3.97** | **3.73** |

### 4.4 Automatic Adaptation Between Fixed- and Variable-Length Tokens

Our model provides two tokenization paths that share the same backbone: a variable-length tokenization path guided by the boundary detector and chunk-level MoE fusion, and a fixed-length tokenization path that partitions the sequence into constant-size chunks. A learned gate $(g_{var}, g_{fix}) \in [0, 1]$ blends the two representations, enabling the model to adaptively balance stable fixed-length tokens with event-synchronous variable-length tokens. As shown in Figure 7, the gate tracks local regime: when the input is stationary (e.g., smooth cycles), $g_{fix} > g_{var}$, steering the model toward the fixed-length path to preserve short-range statistics and reduce variance. In contrast, when the signal exhibits spikes, steps, or abrupt regime changes, $g_{var} > g_{fix}$, shifting weight to the variable-length path, where the boundary head inserts cuts around events and the MoE aggregates event-aware chunk statistics.

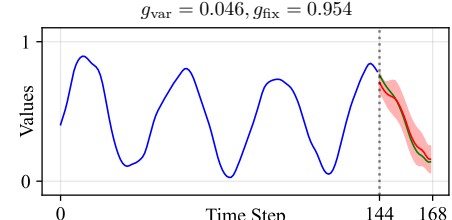

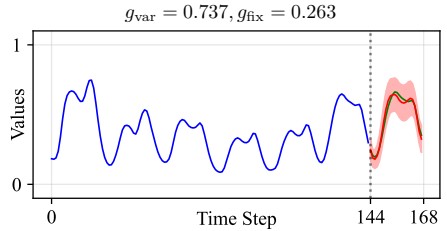

Figure 7: Adaptive tokenization via gating.

## 5 Conclusion, Limitation, and Future Work

We present Boundary-aware Tokenization (BT-LSM), a dynamic chunking framework that detects regime changes and forms event-aligned tokens for time-series forecasting. Across datasets, the advantage of variable-length token emerges where event-driven dynamics occur, whether explicit (alarms, ramps) or implicit (irregular timestamps, hidden bursts). As a result, BT-LSM show consistently improved forecasting accuracy while remaining robust in stationary regimes due to the gating refinement. This confirms our framing as an event-driven forecaster with safety guarantees, rather than just another patching model. Limitations include sensitivity of boundary detection to hyperparameters, added training complexity from gating, and untested robustness under extremely noisy or highly irregular real-world signals; scalability to very long sequences also remains constrained by padding and masking. Future work includes incorporating weak supervision or domain knowledge to guide boundaries, extending to hierarchical or multi-resolution chunking, scaling to foundation-style pretraining, and exploring applications in generative modeling and causal analysis of event-driven systems.

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

## A APPENDIX: USAGE OF LLM

In preparing this work, we leveraged Large Language Models (LLMs) as auxiliary tools for drafting, editing, and refining the manuscript. LLMs assisted in improving clarity, consistency, and presentation of technical content, and also supported organizational tasks such as summarizing experiment logs, formatting tables, and suggesting consistent terminology across sections. Importantly, all scientific ideas, modeling, and experiments remain fully original to the authors; LLMs were not used to design the methodology, but rather to streamline the process of communicating our findings more clearly and efficiently.

## B APPENDIX: ENVIRONMENT SETUP

We evaluate our model and other cutting-edge methods using diverse datasets. Specifically, we have: (1) *Building* Addison et al. (2019). It's a dataset consisting of several buildings' energy consumption and temperature data from the American Society of Heating, Refrigerating and Air-Conditioning Engineers (ASHRAE). (2) *Spain* Kolasniwash (2019). This dataset contains 4 years of electrical consumption, generation, pricing, and weather data for Spain. (3) *Consumption* Fedesoriano (2022). The data consists of $52,416$ observations of energy consumption on a 10-minute window for Tetouan city located in the north of Morocco. Meteorology data (such as weather, temperature, etc.) is also included. (4) *Residential* Sri Polu (2019). The data set contains hourly power usage in kWh from January 2016 to August 2020 in Houston, Texas, USA. A historical weather report is also contained. (5) *Solar* AI Maverick (2023) is a renewable energy (i.e., negative loads) dataset, including solar data, temperature, date information, etc. (6) *ETT* (Electricity Transformer Temperature) datasets Zhou et al. (2021) are collected from two different electric transformers labeled with 1 and 2, and each of them use a resolution of 1 hour denoted with h. Thus, in total we have two ETT datasets: *ETTh1*, and *ETTh2*. (7) *Traffic* dataset Lai et al. (2018) records the road occupancy rates from different sensors on San Francisco freeways. For all datasets, we follow a consistent preprocessing pipeline: sequences are split by weeks, and models are trained to forecast the next $24$ hours based on historical inputs within the same week. This setup ensures comparability across domains while aligning with practical forecasting needs. To further assess robustness, we also evaluate alternative prediction lengths (e.g., 12 and 48 hours) and report these results in the Appendix F.1.

We conducted all experiments on a high-performance computing cluster designed for large-scale AI and time-series research. Compute nodes were equipped with NVIDIA A100 GPUs with 80 GB memory (CUDA 12.7) and dual AMD EPYC 7413 processors with 48 CPU cores at 3.6 GHz, running Linux (kernel 4.18). Each job used GPUs with sufficient memory to accommodate multi-step horizon forecasting and larger model variants. All baselines (PatchTST, GPT, Informer, Contformer, Chronos) and our method were implemented in Python 3.9 with PyTorch. Development and debugging were performed in PyCharm to ensure reproducibility and streamlined iteration.

### B.1 OPTIMIZATION AND SCHEDULE.

We train with AdamW using a learning rate of 1e-4 and weight decay of 1e-5. Global gradient norm clipping is set to 1.0. The learning rate follows Cosine Annealing with Warm Restarts, using an initial restart period of 20 epochs and a multiplier of 2 for subsequent cycles, with a minimum learning rate of 1e-6. Mini-batches of size 16 are used for both training and evaluation. The autoregressive chunk decoder is trained with teacher forcing.

**Training protocol and checkpointing.** Models are trained for 500 epochs. After each epoch, we evaluate on the validation loader and persist the best checkpoint to disk based on the lowest validation loss. Before final evaluation and visualization, we reload the best checkpoint and switch the model to evaluation mode. Autograd anomaly detection is enabled during development for debugging, and gradients are clipped before each optimizer step.

### B.2 BASELINE MODELS.

**ODE-RNN (Rubanova et al., 2019).** We adapt the *ODE-RNN* baseline to our forecasting pipeline. The model uses an LSTMCell encoder with hidden width $d_{\text{model}} = 256$ to process the observed

sequence, followed by continuous-time evolution of the hidden state via a neural ODE solver (Dormand-Prince, `dopri5`) with step size $\Delta t = 0.1$. The ODE function is parameterized by a two-layer MLP with hidden expansion $2 \times d_{\text{model}}$ and `tanh` activation. At each forecast step, the hidden state is integrated forward in continuous time, and two linear heads map it to predictive parameters: a mean head and a log-variance head (output dimension $c_{\text{out}} = 1$). The decoder operates autoregressively, feeding the predicted mean back into the LSTMCell to refine future states. Dropout and attention modules are not used, in line with the original ODE-RNN design. Under this configuration, the ODE-RNN has approximately 2.1M parameters, which is comparable in scale to our Informer baseline.

**RNN-$\Delta t$ (Che et al., 2018).** We consider a recurrent baseline that explicitly incorporates elapsed time between observations. The model augments each input $x_t$ with its corresponding time gap $\Delta t_t$, computed from the observation mask, and processes them jointly through an `LSTMCell` of hidden width $d_{\text{model}} = 256$. Context encoding proceeds step by step over the input window, with masked values replaced by zeros and $\Delta t_t$ tracking the time since last observation. During autoregressive forecasting, the time gap grows deterministically by one unit per step, reflecting the absence of new observations, and is concatenated with the previous prediction before being fed back into the `LSTMCell`. Two nonlinear heads project the hidden state into per-step Gaussian parameters (mean and log-variance, $c_{\text{out}} = 1$), enabling Gaussian NLL training as in our other baselines. No attention or convolutional modules are used. This RNN-$\Delta t$ baseline has roughly 2.0M parameters, making it comparable in size to the Informer and ODE-RNN models.

**InformerZhou et al. (2021).** We use the official Informer implementation (Zhou et al., 2021) as an encoder–decoder Transformer with ProbSparse self-attention, value+positional+time ("timeF", hourly) embeddings, and the standard distilling pathway (1D convolution + activation + max-pool) between encoder blocks to halve the sequence length before deeper layers. Our configuration mirrors common practice and our code wrapper: encoder depth $e_{\text{layers}} = 2$, decoder depth $d_{\text{layers}} = 1$, model width $d_{\text{model}} = 256$, $n_{\text{heads}} = 8$, feed-forward width $d_{\text{ff}} = 512$, GELU activations, and dropout $= 0.1$. The decoder consumes the last `label_len` observations concatenated with `horizon` zero placeholders, and a linear projection maps each forecast step to two channels (mean and log-variance; `c_out`=2), enabling Gaussian NLL training consistent with our baseline interface. Under this setup, the Informer baseline has roughly 2.05M parameters in our experiments.

**ContiFormer Chen et al. (2023).** ContiFormer extends Transformers to irregularly sampled data by working directly in continuous time. Given observations $\{(x_i, t_i)\}$, each layer constructs *continuous* key/value trajectories by evolving latent states with an ODE between events, while the query is a time-continuous function $q(t)$ obtained by spline interpolation through the discrete query embeddings. The core module is continuous-time multi-head attention (CT-MHA), which replaces the discrete dot-product with a scaled inner product $\int q(\tau) \cdot k(\tau) \, d\tau$ over time, producing representations at arbitrary timestamps. A reparameterization yields a parallelizable implementation that preserves Transformer-style layer norms and position-wise feed-forward blocks. In our experiments, we use a lightweight encoder-decoder stack with $e_{\text{layers}} = 2$ and $d_{\text{layers}} = 1$, hidden width $d_{\text{model}} = 256$, and $n_{\text{heads}} = 8$, resulting in approximately 2.1M trainable parameters. While ContiFormer is designed for irregular sequences with missing or asynchronous measurements, our datasets are sampled at regular hourly intervals; thus, the irregularity-aware modules are not strictly necessary but still provide a principled mechanism for continuous-time forecasting. This design highlights how continuous-time attention can unify sequence modeling across both regularly and irregularly sampled domains.

**PatchTST (Nie et al., 2022).** PatchTST is a Transformer-based architecture for time series forecasting that tokenizes inputs into non-overlapping patches and applies self-attention over patch embeddings. We consider two usage modes. In the **zero-shot** setting, we load the official pretrained weights (`ibm-granite/granite-timeseries-patchtst`) and directly apply the model to our datasets without gradient updates. This tests the out-of-distribution generalization ability of pretrained time-series Transformers. In the **fine-tuned** setting, we adapt the pretrained model to each dataset using our training pipeline. We explore multiple finetuning strategies: updating all weights (*full*), training only the regression head (*head*), or introducing a lightweight mean–variance calibrator (*adapter*). Unless otherwise noted, we report results with the full finetuning strategy. The forecaster outputs per-step Gaussian parameters (mean and log-variance, $c_{\text{out}} = 1$) for consis-

tency with our baseline interface. The fine-tuned model contains roughly 20M parameters, making it significantly larger than our recurrent baselines but still feasible to train within our experimental setup.

**Chronos (Ansari et al., 2024).** Chronos is a pretrained family of sequence-to-sequence forecasters based on Transformer backbones T5 (`amazon/chronos-t5-small`). The model outputs predictive quantiles, which we convert into Gaussian parameters (mean and log-variance) for consistency with our evaluation framework. We adopt the **fine-tuned** setting, where a lightweight dataset-specific calibrator rescales the predicted mean and shifts the variance estimates while keeping the pretrained Chronos backbone frozen. This provides efficient adaptation with negligible computational overhead and improves dataset alignment. Chronos models typically contain tens of millions of parameters (depending on the variant), but the calibration layer adds only a few hundred additional parameters, making fine-tuning fast and stable.

**GPT (Radford et al., 2019).** We adapt the original *GPT* architecture for time series forecasting, preserving the exact structure used in language modeling. Specifically, we employ a stack of masked self-attention blocks with residual connections, layer normalization, and GELU activations, identical to GPT-2. The model projects scalar time-series inputs into the embedding space, adds learnable positional embeddings, and autoregressively generates the next horizon of values. At each step, a linear head outputs Gaussian parameters (mean and log-variance) for training with negative log-likelihood. We experiment with multiple GPT-2 configurations ranging from small variants ($\sim$10M parameters) up to the 355M parameters. This setup ensures that our results reflect the direct transfer of GPT-style Transformer architectures to the time-series domain, without introducing architectural modifications.

### B.3 BT-LSM MODEL CONFIGURATION.

Unless otherwise stated, we instantiate BT-LSM with model width 128, forecasting horizon 24 steps, and a maximum of 50 chunks per sequence. Raw inputs of dimension 1 are linearly projected to the model width and combined with positional encodings, followed by a gated feed-forward encoder. Chunk boundaries are produced by an adaptive detector in *hybrid* mode with a boundary ratio of 0.05 and a minimum chunk length of 3. Per-chunk representations are formed by a Mixture-of-Experts layer with five experts (Mean, First-Token, Max, Min, and Attention) under soft gating with temperature 1.0. Inter-chunk temporal dependencies are modeled with a GPT-2 causal decoder configured as `gpt2-tiny` and a context budget equal to the maximum chunk count. An exponential-moving-average smoother (alpha 0.3) refines decoded chunk embeddings. A cross-attention upsampler with 8 heads maps chunk-level latents back to horizon-length latents. The prediction head outputs both mean and log-variance for heteroscedastic forecasting.

BT-LSM introduces adaptive tokenization into Transformer-style forecasters by segmenting the input sequence into variable-length chunks. The input sequence $\mathbf{x} \in \mathbb{R}^{L \times d_{\text{in}}}$ is first embedded by a linear projection ($d_{\text{in}} \rightarrow d_{\text{model}}$) and sinusoidal positional encoding, followed by a two-layer feed-forward encoder with hidden size $2d_{\text{model}}$ and GELU activations. Unless otherwise stated, we use $d_{\text{model}} = 128$.

A dedicated adaptive **Boundary Detector** then selects chunk boundaries. This module employs three parallel temporal convolutions with kernel sizes $\{3, 5, 7\}$ and output width $d_{\text{model}}/2$, whose features are concatenated and projected through a two-layer perceptron to yield boundary scores. Learned scores are combined with normalized velocity, curvature, and energy-change cues using three trainable scalar weights. A ratio constraint ensures approximately $\lfloor L \cdot \rho \rfloor$ boundaries (with $\rho = 0.07$ by default), subject to a minimum spacing of 3–5 steps. The output consists of soft probabilities (for visualization) and hard binary masks (for chunk construction).

Boundaries are converted into chunk-level tensors by the chunk embedding, which collects subsequences into padded arrays $[B, M_{\text{max}}, \ell_{\text{max}}, d_{\text{model}}]$ and records lengths, masks, and spans. Each chunk is then compressed into a single vector by the chunking MoE. The MoE consists of five parallel experts: (i) mean pooling, (ii) max pooling, (iii) min pooling, (iv) first-token selection, and (v) single-query attention with $h = 4$ heads. Expert outputs are mixed by a router that maps simple chunk statistics (mean, first, last) through a two-layer perceptron, producing a softmax distribution over experts.

Chunk embeddings are processed autoregressively by a causal decoder with $N = 6$ self-attention layers, each containing $h = 8$ heads and hidden size $d_{\text{model}} = 128$. Residual connections, layer normalization, and two-layer feed-forward sub-blocks (hidden size $4d_{\text{model}}$) follow the standard Transformer design. To reduce high-frequency instability between adjacent chunks, we apply a lightweight **Chunking Smoothing** layer. This module maintains a running exponential moving average with a learnable smoothing factor $\alpha \in (0, 1)$, applied sequentially across valid chunk positions, while leaving padded slots untouched.

For horizon forecasting, we employ a **CrossAttention Decoder**. It maintains a bank of $H$ learnable queries (one per prediction step), augmented with sinusoidal positional encodings. Queries attend to the memory of chunk embeddings via multi-head cross-attention ($h = 8$, hidden size $d_{\text{model}} = 128$), followed by a two-layer feed-forward refinement block. This stage produces decoded latents of shape $[B, H, d_{\text{model}}]$ and attention maps $[B, H, M_{\text{max}}]$.

In the end, it maps each decoded latent into probabilistic forecasts. Two parallel three-layer MLP heads (hidden size $d_{\text{model}}/2$ with GELU activations) output the predictive mean and log-variance, respectively. Log-variance is clamped to $[-10, 2]$ for numerical stability.

The complete model integrates embedding, boundary detection, chunk creation, MoE compression, autoregressive decoding, EMA smoothing, cross-attentive upsampling, and probabilistic prediction. In its standard configuration ($d_{\text{model}} = 128$, $N = 6$ decoder layers, $h = 8$). Importantly, unlike plain Transformers, the architecture always incorporates adaptive boundary detection and expert-guided chunk compression, ensuring that forecasts are structured by dynamically learned temporal units rather than fixed-length patches.

## C ALGORITHM OF BT-LSM

---

**Algorithm 1** The Training of Boundary-aware Tokenization Large Signal Model (BT-LSM)

---

**Require:** Observed sequence $\mathbf{x}_{1:T}$, horizon $H$, target boundary-rate $1/\kappa$, hyperpaparameters $\lambda_{\text{boundary}}, \lambda_{\text{cont}}$

**Ensure:** Predictions $\hat{\mathbf{x}}_{T+1:T+H}$, loss $\mathcal{L}$

1: **Embed:** $\mathbf{e}_{1:T} \leftarrow \text{LinearProj}(\mathbf{x}_{1:T})$
2: **Boundary probs:** $p_{1:T-1} \leftarrow \sigma\big(\mathbf{w}_v^\top (\Delta\mathbf{e}) + \mathbf{w}_a^\top (\Delta^2\mathbf{e}) + \mathbf{w}_{\mathcal{E}}^\top (\Delta\|\mathbf{e}\|)\big)$
3: **Hard boundaries (STE):** $b_1 \leftarrow 1$; $b_t \leftarrow \mathbb{1}\{p_t \geq 0.5\}$ (backprop: $\partial b_t/\partial p_t \approx 1$)
4: **Chunks:** get start indices $\{t^{(m)}\}_{m=1}^M$ from $\{b_t\}$; set $t^{(M+1)} \leftarrow T$
5: **MoE fuse per chunk:** for each $m$, compute mean, boundary, max-norm, attention pools; gate and fuse $\rightarrow \mathbf{z}_m^{\text{MoE}}$
6: **Add metadata:** $\ell_m \leftarrow t^{(m+1)} - t^{(m)}$; $\mathbf{z}_m \leftarrow [\,\mathbf{z}_m^{\text{MoE}} \| \ell_m \| t^{(m)}\,]$
7: **(Batching)** Pad $\{\mathbf{z}_m\}$ across sequences; build masks
8: **Chunk Transformer:** $\hat{\mathbf{z}}_{1:M} \leftarrow \text{CausalDecoderBlocks}(\{\mathbf{z}_m\})$
9: **Smoothing (EMA):** $\bar{\mathbf{z}}_1 \leftarrow \hat{\mathbf{z}}_1$; $\bar{\mathbf{z}}_m \leftarrow (1-\tilde{p}_m)\bar{\mathbf{z}}_{m-1} + \tilde{p}_m\hat{\mathbf{z}}_m$
10: **Gating refinement:** fuse $\bar{\mathbf{z}}_m^{\text{var}}, \bar{\mathbf{z}}_m^{\text{fix}} \rightarrow \bar{\mathbf{z}}_m$
11: **Cross-attention decode:** for $\tau = T+1:T+H$,
    $\mathbf{q}_\tau \leftarrow \text{FourierEmbed}(\tau)$; $\mathbf{h}_\tau \leftarrow \text{MHA}(\mathbf{q}_\tau, \bar{\mathbf{z}}_{1:M}, \bar{\mathbf{z}}_{1:M})$; $\hat{\mathbf{x}}_\tau \leftarrow \text{FFN}(\mathbf{h}_\tau)$
12: **Loss:** $\mathcal{L} \leftarrow \frac{1}{H} \sum_{h=1}^H \|\mathbf{x}_{T+h} - \hat{\mathbf{x}}_{T+h}\|_2^2 + \lambda_{\text{boundary}}\left(\frac{1}{T}\sum_{t=1}^T b_t - \frac{1}{\kappa}\right)^2 + \lambda_{\text{cont}}\|\hat{\mathbf{x}}_{T+1} - \mathbf{x}_T\|_2^2$

---

# D DISCUSSION AND PROOF OF THEOREM 1

**Time warp.** Consider a (strictly) increasing, continuously differentiable bijection $\phi_m$ : $[t^{(m)}, t^{(m+1)}] \to [t^{(m)}, t^{(m+1)}]$ with $\phi_m(t^{(m)}) = t^{(m)}$ and $\phi_m(t^{(m+1)}) = t^{(m+1)}$. Define the time-warped signal and embedding within the chunk by $\tilde{\mathbf{x}}(t) := \mathbf{x}(\phi_m(t))$ and $\tilde{\mathbf{e}}(t) := \mathbf{e}(\phi_m(t))$. Boundaries and chunk endpoints are unchanged by construction.

**Assumption 1** (No intra-chunk position features). *Within-chunk experts depend only on $\mathbf{e}(t)$ (content) and not on the absolute index $t$ (except for the designated boundary expert at $t^{(m)}$).*

**Assumption 2** (Bounded embeddings). *$\mathbf{e}(t)$ is measurable and bounded on $[t^{(m)}, t^{(m+1)}]$; $\alpha(t) = \exp(\mathbf{w}^\top \mathbf{e}(t))$ is integrable and strictly positive.*

**Theorem 1** (Invariance to intra-chunk resampling). *Under Assumptions 1-2, for any monotone time warp $\phi_m$ as above, each expert output is invariant:*

$$\tilde{\mathbf{z}}_m^{\mathrm{mean}} = \mathbf{z}_m^{\mathrm{mean}}, \quad \tilde{\mathbf{z}}_m^{\mathrm{boundary}} = \mathbf{z}_m^{\mathrm{boundary}}, \quad \tilde{\mathbf{z}}_m^{\mathrm{max}} = \mathbf{z}_m^{\mathrm{max}}, \quad \tilde{\mathbf{z}}_m^{\mathrm{attn}} = \mathbf{z}_m^{\mathrm{attn}}.$$

*Consequently, $\tilde{\mathbf{z}}_m^{\mathrm{MoE}} = \mathbf{z}_m^{\mathrm{MoE}}$ and the concatenated token $\tilde{\mathbf{z}}_m = [\tilde{\mathbf{z}}_m^{\mathrm{MoE}} \| L_m \| t^{(m)}]$ equals $\mathbf{z}_m$. Therefore the full model outputs are invariant: $\tilde{\bar{\mathbf{z}}}_m = \bar{\mathbf{z}}_m$ for all $m$, and $\hat{\tilde{\mathbf{x}}}_{T+1:T+H} = \hat{\mathbf{x}}_{T+1:T+H}$.*

*Proof.* We prove expertwise invariance.

*Mean.* By change of variables $u = \phi_m(t)$ with $du = \phi_m'(t)\, dt$ and $\phi_m$ bijective,

$$\tilde{\mathbf{z}}_m^{\mathrm{mean}} = \frac{1}{L_m} \int_{t^{(m)}}^{t^{(m+1)}} \tilde{\mathbf{e}}(t)\, dt = \frac{1}{L_m} \int_{t^{(m)}}^{t^{(m+1)}} \mathbf{e}(u)\, \frac{du}{\phi_m'(\phi_m^{-1}(u))} \overset{(\star)}{=} \frac{1}{L_m} \int_{t^{(m)}}^{t^{(m+1)}} \mathbf{e}(u)\, du = \mathbf{z}_m^{\mathrm{mean}},$$

where $(\star)$ uses the standard change-of-variables identity: integrating $\mathbf{e}(\phi_m(t))$ w.r.t. $dt$ is identical to integrating $\mathbf{e}(u)$ w.r.t. $du$ over the same interval; the Jacobian cancels because the bounds map to the same endpoints.

*Boundary.* $\tilde{\mathbf{z}}_m^{\mathrm{boundary}} = \tilde{\mathbf{e}}(t^{(m)}) = \mathbf{e}(\phi_m(t^{(m)})) = \mathbf{e}(t^{(m)}) = \mathbf{z}_m^{\mathrm{boundary}}$.

*Max-norm exemplar.* The map $t \mapsto \|\tilde{\mathbf{e}}(t)\|_2 = \|\mathbf{e}(\phi_m(t))\|_2$ is a reparameterization of the same curve $\{\|\mathbf{e}(u)\|_2 : u \in [t^{(m)}, t^{(m+1)}]\}$; hence its maximum value and argmax point (pulled back by $\phi_m^{-1}$) are unchanged. Thus the selected embedding equals the original maximizer's embedding, so $\tilde{\mathbf{z}}_m^{\mathrm{max}} = \mathbf{z}_m^{\mathrm{max}}$.

*Attention.* Numerator and denominator transform with the same Jacobian:

$$\tilde{\mathbf{z}}_m^{\mathrm{attn}} = \frac{\int \exp(\mathbf{w}^\top \mathbf{e}(\phi_m(t)))\, \mathbf{e}(\phi_m(t))\, dt}{\int \exp(\mathbf{w}^\top \mathbf{e}(\phi_m(t)))\, dt} = \frac{\int \exp(\mathbf{w}^\top \mathbf{e}(u))\, \mathbf{e}(u)\, du}{\int \exp(\mathbf{w}^\top \mathbf{e}(u))\, du} = \mathbf{z}_m^{\mathrm{attn}}.$$

Thus all experts are invariant. MoE fusion with fixed gates preserves equality. The appended metadata $(L_m, t^{(m)})$ is boundary-defined and thus unchanged. Hence $\tilde{\mathbf{z}}_m = \mathbf{z}_m$ for all $m$.

Downstream, the chunk Transformer and EMA smoothing are functions of the sequence $\{\mathbf{z}_m\}$ only (with causal masks), so their outputs coincide: $\tilde{\bar{\mathbf{z}}}_m = \bar{\mathbf{z}}_m$. Finally, the cross-attention decoder uses identical queries (Fourier embeddings of $T+1:T+H$) and identical keys/values $\{\bar{\mathbf{z}}_m\}$, hence produces the same $\hat{\mathbf{x}}_{T+1:T+H}$. □

**Remark 1** (What would break invariance?). *Including intra-chunk positional features (e.g., per-step sinusoidal encodings inside experts), or using density-dependent normalizations not cancelling under change of variables, would violate Assumption 1 and can break invariance.*

**Corollary 1** (Discrete implementation: robustness and rate). *Suppose each chunk $m$ is sampled on grids $\{t_i\}$ and $\{\tilde{t}_j\}$ before and after a time warp, with maximal spacings $\Delta_m$ and $\tilde{\Delta}_m$. If $\mathbf{e}(t)$ is Lipschitz on $[t^{(m)}, t^{(m+1)}]$ and attention scores are bounded, then the discrete Riemann-sum implementations of the four experts are* consistent *and satisfy*

$$\|\tilde{\mathbf{z}}_m^k - \mathbf{z}_m^k\| \leq C_k \cdot \max\{\Delta_m, \tilde{\Delta}_m\}, \qquad k \in \{\mathit{mean, boundary, max, attn}\},$$

*for constants $C_k$ depending on Lipschitz and curvature bounds. Consequently, the end-to-end outputs satisfy $\|\hat{\tilde{\mathbf{x}}}_{T+1:T+H} - \hat{\mathbf{x}}_{T+1:T+H}\| \leq \mathcal{O}(\max_m \max\{\Delta_m, \tilde{\Delta}_m\})$.*

**Implications.** The result shows BT-LSM is exactly invariant to any monotone reparameterization *within* chunks (in the continuous formulation) and is robust in the discrete setting with an error controlled by sampling resolution. Practically, as long as (i) boundaries are stable, (ii) experts use content-only statistics (no intra-chunk positional encodings), and (iii) attention is normalized, forecasts are insensitive to intra-chunk resampling/warping.

# E ADDITIONAL EXAMPLES OF FORECASTING RESULTS

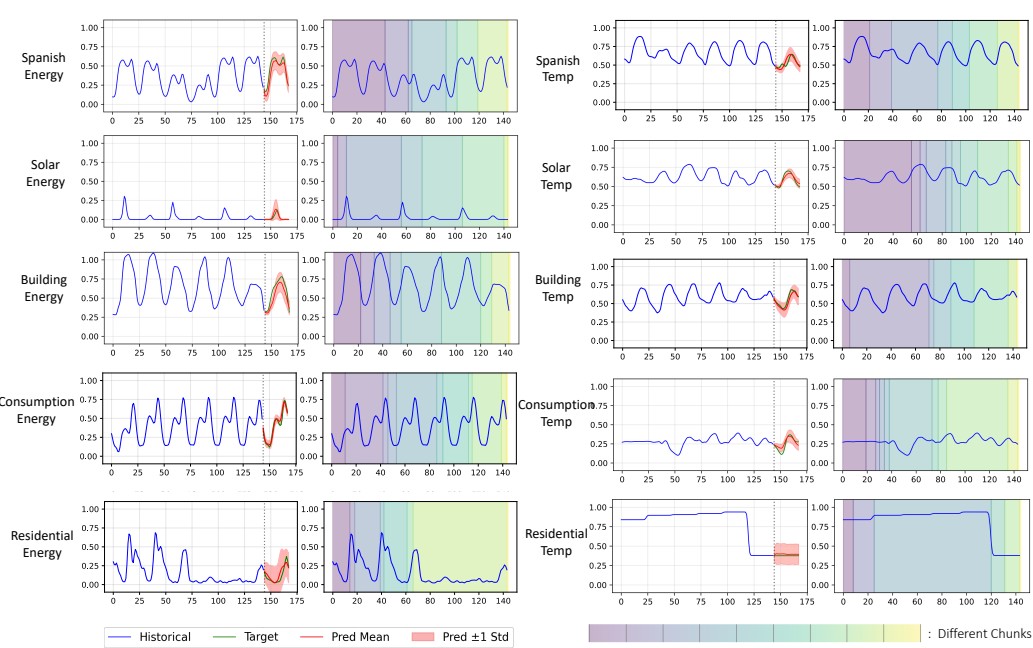

Figure 8: Qualitative results on five datasets (left block) and their Temperature counterparts (right block). Boundaries align with inflections, spikes, or regime changes, while plateaus yield long, stable chunks.

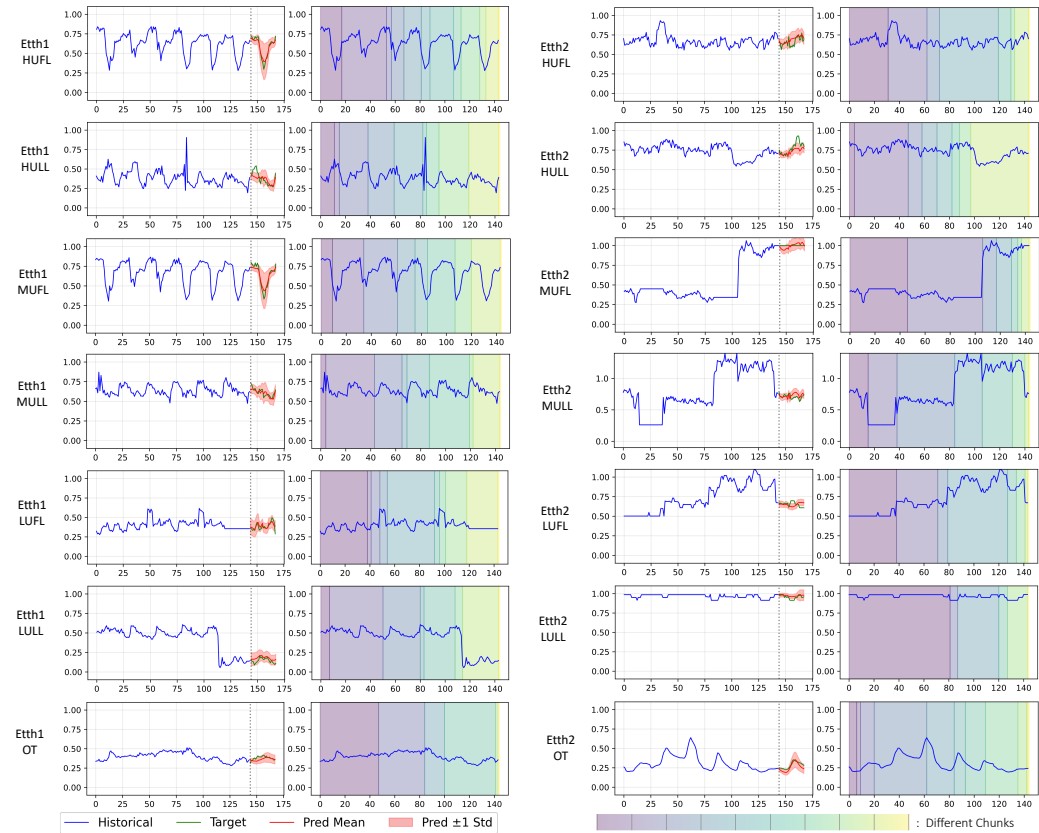

Figure 9: ETT benchmark breakdown. For each target in **ETTh1** and **ETTh2** (HUFL, HULL, MUFL, MULL, LUFL, LULL, OT). The left panel shows the forecast (mean in red with ±1 std; blue historical; green target) and the right panel shows the learned chunk segmentation (shaded). The model places boundaries at step changes and sharp transitions, yielding event-synchronous tokens.

## F   COMPLETE MSE RECORDS OF ETT DATASET

Table 3: Performance comparison on ETTh1 and ETTh2 datasets (all values in $\times 10^{-3}$).

| Method | #Para | ETTh1 | | | | | | | ETTh2 | | | | | | |
|---|---|---|---|---|---|---|---|---|---|---|---|---|---|---|---|
| | | HUFL | HULL | MUFL | MULL | LUFL | LULL | OT | HUFL | HULL | MUFL | MULL | LUFL | LULL | OT |
| RNN-$\Delta t$ | 0.33M | 20.04 ± 2.28 | 10.82 ± 1.22 | 21.43 ± 2.42 | 10.21 ± 1.12 | 16.53 ± 1.83 | 7.24 ± 0.81 | 3.52 ± 0.38 | 7.21 ± 0.79 | 8.12 ± 0.90 | 14.83 ± 1.63 | 4.46 ± 0.09 | 3.13 ± 0.34 | 0.86 ± 0.09 | 5.23 ± 0.57 |
| ODE-RNN | 0.53M | 22.11 ± 2.46 | 11.63 ± 1.28 | 23.18 ± 2.57 | 11.84 ± 1.30 | 17.92 ± 1.98 | 7.93 ± 0.87 | 3.89 ± 0.42 | 7.94 ± 0.86 | 8.92 ± 0.97 | 16.34 ± 1.81 | 5.81 ± 0.12 | 3.47 ± 0.38 | 0.94 ± 0.10 | 5.94 ± 0.66 |
| Informer | 2.05M | 18.72 ± 2.05 | 9.41 ± 1.04 | 18.71 ± 2.04 | 9.82 ± 1.08 | 14.31 ± 1.56 | 6.27 ± 0.70 | 3.21 ± 0.35 | 10.54 ± 1.19 | 11.17 ± 1.26 | 21.03 ± 2.36 | 8.51 ± 0.18 | 4.92 ± 0.54 | 1.31 ± 0.14 | 8.21 ± 0.90 |
| ContiFormer | 1.85M | 16.43 ± 1.80 | 8.54 ± 0.94 | 16.42 ± 1.78 | 8.63 ± 0.94 | 12.74 ± 1.38 | 5.59 ± 0.62 | 2.84 ± 0.31 | 9.07 ± 1.01 | 9.82 ± 1.09 | 18.41 ± 2.02 | 7.36 ± 0.15 | 4.17 ± 0.46 | 1.08 ± 0.12 | 7.10 ± 0.78 |
| GPT | 19.05M | 15.47 ± 1.26 | 7.92 ± 0.67 | 15.31 ± 1.24 | 7.83 ± 0.65 | 11.93 ± 0.98 | 5.02 ± 0.41 | 2.61 ± 0.22 | 5.62 ± 0.48 | 6.41 ± 0.55 | 11.62 ± 0.97 | 4.16 ± 0.08 | 2.41 ± 0.21 | 0.66 ± 0.05 | 4.23 ± 0.36 |
| Chronos | 46.15M | 16.52 ± 1.40 | 8.12 ± 0.69 | 16.08 ± 1.36 | 8.24 ± 0.71 | 12.31 ± 1.05 | 5.21 ± 0.44 | 2.72 ± 0.24 | 6.29 ± 0.53 | 7.02 ± 0.60 | 12.53 ± 1.07 | 6.10 ± 0.12 | 2.73 ± 0.24 | 0.73 ± 0.06 | 4.82 ± 0.41 |
| PatchTST | 0.61M | 15.82 ± 1.33 | 8.01 ± 0.68 | 15.72 ± 1.32 | 8.03 ± 0.68 | 12.10 ± 1.03 | 5.08 ± 0.43 | 2.65 ± 0.23 | 7.11 ± 0.62 | 7.63 ± 0.67 | 13.73 ± 1.21 | 7.09 ± 0.14 | 3.01 ± 0.27 | 0.80 ± 0.07 | 5.31 ± 0.47 |
| Ours | 0.90M | 13.90 ± 0.38 | 7.56 ± 0.32 | 14.38 ± 0.42 | 7.14 ± 0.29 | 11.31 ± 0.36 | 4.79 ± 0.16 | 2.40 ± 0.12 | 5.15 ± 0.19 | 5.96 ± 0.21 | 10.54 ± 0.40 | 3.94 ± 0.17 | 2.05 ± 0.08 | 0.59 ± 0.03 | 3.73 ± 0.16 |

### F.1   MULTI-HORIZON FORECASTING RESULTS

To evaluate robustness across different forecasting horizons, we extend our experiments on the ETT benchmark to include prediction lengths of 12, 24, and 48 steps. Table 4 reports the mean squared error (MSE $\times 10^{-3}$) across horizons. We observe that while all models degrade with longer horizons, our BT-LSM consistently maintains lower error. At shorter horizons (12 steps), all strong baselines perform competitively, but as the horizon increases to 48 steps, the performance gap widens, with BT-LSM showing the best stability. This confirms that boundary-aware tokenization not only aligns with event-driven structure but also offers robustness across varying prediction lengths, a critical property in practical forecasting scenarios.

Table 4: Multi-horizon forecasting results on ETTh1 and ETTh2 (MSE $\times 10^{-3}$).

| Method | ETTh1 | | | ETTh2 | | |
|---|---|---|---|---|---|---|
| | H=12 | H=24 | H=48 | H=12 | H=24 | H=48 |
| RNN-$\Delta$t | 9.62 | 17.92 | 24.34 | 10.15 | 21.86 | 29.41 |
| ODE-RNN | 10.21 | 20.89 | 27.73 | 9.92 | 19.40 | 26.62 |
| PatchTST | 5.54 | 9.57 | 14.41 | 5.22 | 12.99 | 17.84 |
| ContiFormer | 5.12 | 9.13 | 13.56 | 5.05 | 16.23 | 21.32 |
| Chronos | 4.91 | 10.63 | 9.14 | 4.84 | 12.12 | 17.67 |
| BT-LSM (ours) | **4.37** | **5.97** | **8.48** | **4.29** | **11.12** | **15.06** |

# G TRAINING AND INFERENCE EFFICIENCY

Besides forecasting accuracy, computational efficiency is also important for practical deployment. We report both the average training time per epoch and the inference latency per test sequence (in milliseconds). As shown in Table 5, our BT-LSM achieves lower inference latency than large baselines such as Chronos and GPT, while remaining competitive in training time compared to lightweight fixed-length models. This demonstrates that boundary-aware chunking not only improves accuracy but also reduces compute overhead.

Table 5: Training and inference efficiency comparison. Training time measured per epoch; inference latency per sequence (ms).

| Method | #Params (M) | Train Time (s/epoch) | Inference Latency (ms) |
|---|---|---|---|
| RNN-$\Delta$t | 0.33 | 12.4 | 2.3 |
| ODE-RNN | 0.53 | 18.7 | 4.1 |
| PatchTST | 0.61 | 26.5 | 8.9 |
| ContiFormer | 1.85 | 34.2 | 7.3 |
| GPT | 19.05 | 112.6 | 20.5 |
| Chronos | 46.15 | 158.3 | 24.7 |
| BT-LSM (ours) | 0.90 | 29.8 | 4.7 |

## G.1 ZERO-SHOT GENERALIZATION TO TRAFFIC DATA

To further evaluate generalization ability, we conduct a zero-shot experiment: models are pretrained on all datasets *except Traffic*, and then directly evaluated on the Traffic dataset without fine-tuning. This setting simulates practical scenarios where models are deployed in unseen domains. Table 6 reports the mean squared error (MSE $\times 10^{-3}$) across model scales. We find that larger models generally achieve better zero-shot accuracy, but our BT-LSM maintains strong performance even at small scale (0.9M parameters), outperforming conventional baselines of similar size.

Table 6: Zero-shot forecasting on Traffic dataset (pretrained on other datasets). MSE reported in $\times 10^{-3}$.

| Model Size of BT-LSM | Params (M) | Zero-Shot MSE |
|---|---|---|
| Small | 0.9 | 12.3 |
| Medium | 10 | 9.8 |
| Large | 50 | 7.6 |
| XL | 124 | 6.1 |
| XXL | 355 | **5.4** |

