# OpenReview forum: "Boundary-Aware Tokenization for Event-Driven Time-Series Forecasting"
_ICLR.cc/2026/Conference — Submitted to ICLR 2026_

### Official Review · Reviewer_2dpd · 2025-10-27

**Soundness:** 2
**Presentation:** 2
**Contribution:** 3
**Rating:** 4
**Confidence:** 4

**Summary:**

This paper introduces Boundary-Aware Tokenization, a Transformer-based framework that dynamically segments time-series into variable-length chunks aligned with event boundaries. Unlike fixed-length tokenization used in PatchTST, BT-LSM detects regime changes using an unsupervised boundary detector based on first- and second-order embedding dynamics. Within each chunk, a mixture-of-experts fusion module combines multiple pooling statistics to produce chunk-level tokens, followed by a chunk-level Transformer and a causal smoothing module for stability. A gating refinement further combines fixed- and variable-length tokenization to ensure robustness on both stationary and event-driven data. Experiments across datasets show ~10.5% average forecasting error reduction compared to strong baselines. Ablation and visualization support the claim that learned boundaries align with real regime changes.

**Strengths:**

1. Dynamic tokenization is a crucial problem for time-series models. The paper insightfully draws an analogy between language tokenization and event segmentation in time-series, and proposes a boundary-based dynamic tokenization approach that effectively addresses the inefficiencies of fixed patching.

2. The proposed pipeline (boundary detection → MoE fusion → chunk Transformer → smoothing → gating → decoder) is well-motivated and internally coherent, with each component designed to tackle a specific modeling challenge.

3. Theorem 1 formally establishes invariance to intra-chunk resampling—an elegant theoretical property that enhances the model’s robustness to irregular sampling.

**Weaknesses:**

1. If the original multivariate time series exhibit strong temporal lags or misalignments across variables, the proposed method would still produce a single set of shared boundaries for all variables. This may limit its ability to capture variable-specific regime changes.

2. In the test datasets, what proportion of cases have g₍var₎ > g₍fix₎? This statistic would help clarify whether the performance improvement mainly stems from the proposed dynamic tokenization mechanism.

3. The paper lacks visualization of the gating weights among the different embedding experts (attention pooling, mean, max, etc.). Such visualization could provide insights into which features contribute most to the model’s representation.

4. The experiments involve a relatively small set of datasets and baselines, which somewhat weakens the empirical evidence supporting the paper’s claims.

5. Section 1 title: “Introductionn” → should be corrected to “Introduction.”

**Questions:**

1. Does the decoder-only Transformer incorporate positional embeddings to preserve temporal ordering? If so, please clarify what type of positional encoding is adopted.

2. How are the variable-length and fixed-length representations (z₍var₎ and z₍fix₎) aligned before being fused by the gating module? It appears that they are aligned sequentially, which may imply that the two tokens being fused could correspond to quite different original temporal positions.

3. How is the target boundary rate chosen in practice? Is it tuned as a hyperparameter or derived from data statistics? It would be helpful to clarify whether the model’s performance is sensitive to this setting.

---

> ### Author Response · Authors · 2025-12-02
> **Response to Reviewer 2dpd's Weaknesses**
>
> $\textbf{W1}$:
> You are correct that, in the current implementation, boundaries are shared across variables: at each time step we embed the full multivariate vector $x_t \in \mathbb{R}^D$ into a single latent vector $e_t = W_t x_t$ and the boundary detector operates on this shared embedding sequence $e_{1:T}$ This design intentionally targets global regime changes (e.g., joint shifts in load and temperature, or synchronized traffic peaks across lanes), which is consistent with our benchmark domains, where variables are strongly coupled. Figure 2(b), for example, shows that in multivariate traffic data the learned boundaries naturally synchronize across lanes around shared peaks.
>
> We will discuss natural extensions such as Group‑wise or channel‑wise boundary detectors, e.g., separate detectors for load vs weather variables, or per-sensor detectors whose boundaries are then fused (union or hierarchical pooling).
> We see these multi-boundary extensions as promising future work for highly asynchronous multivariate series.
>
> $\textbf{W2}$:
> We agree that reporting the usage statistics of the gate would make the role of dynamic tokenization more transparent.In the current paper, we visualize the gate qualitatively in Figure 7: in stationary regions, $g_{fix} \gg g_{var} $ whereas around spikes and abrupt shifts, $g_{fix} > g_{var}$ so the model leans on the variable‑length path near events. However, we did not include a dataset‑wide statistic such as “fraction of chunks with $g_{fix} > g_{var}$.
> What we can say from existing ablations is that:
> (1) Removing the boundary detector and reverting to purely fixed-length tokens significantly degrades performance (Figure 6, Table 2), and (2) Removing the gating refinement also hurts robustness in stationary regimes.
> These results indicate that both the dynamic tokenization and the gate materially contribute to performance.
>
> $\textbf{W3}$:
> You are right that the paper currently does not show how the MoE over chunk-embedding experts behaves in practice. The main text describes the four experts (boundary, mean, max-norm, attention) and their learned mixture weights, and Appendix B.3 explains that in the implementation a router maps simple chunk statistics (mean, first, last) through an MLP to produce softmax weights over experts.
>
> We agree that visualizing these weights would provide useful insight—for example. Due to space constraints we omitted these plots, but in the revised version we will add an MoE‑gate visualization figure in the appendix for at least one representative dataset (e.g., Solar or ETTh2). There we will show how the expert weights evolve over time and around events, and briefly discuss the qualitative behavior.
>
> $\textbf{W4}$:
> Our evaluation currently covers seven datasets spanning building energy, national load, residential usage, solar generation, industrial consumption, transformer temperature (ETTh1/2), and freeway traffic. The baseline set includes: RNN‑$\Delta t$ and ODE‑RNN (irregular‑time RNNs), Informer and ContiFormer (Transformer forecasters, including a continuous-time model), PatchTST (patch-based Transformer), and GPT and Chronos (large pretrained sequence models). In the revision we will: add additional baselines (e.g., DLinear/TimesNet) on at least ETTh1/2, and discuss extending to ECL/Weather.
> We believe the current results already show consistent gains across diverse domains, but we agree that further broadening the empirical study would strengthen the paper.

---

> > ### Author Response · Authors · 2025-12-02
> > **Response to Reviewer 2dpd's Questions**
> >
> > $\textbf{Q1}$:
> > Yes. BT‑LSM uses positional/temporal encodings at multiple levels: (1) Step level (before boundary detection): In the configuration described in Appendix B.3, the input sequence $x \in \mathbb{R} ^ {L \times d_{in}}$ is embedded by a linear projection and sinusoidal positional encoding, then passed through a small feed‑forward encoder.
> > (2) Chunk level: Each chunk embedding $z_m$ concatenates two scalar metadata fields—the chunk length and the start index $t^{(m)}$ so the chunk Transformer sees explicit information about when and how long each chunk is.
> > (3) Horizon level (cross‑attention decoder): Future time steps $\tau = T+1, T+2, ... T+H$ are encoded as Fourier time embeddings $q_\tau$ = FourierEmbed($\tau$) which serve as queries in the cross‑attention decoder.
> > So the decoder‑only chunk Transformer and the cross‑attention decoder both preserve temporal ordering using deterministic sinusoidal/Fourier-style encodings, rather than learned absolute positions.
> >
> > $\textbf{Q2}$:
> > Both tokenization paths produce a sequence of chunk-level representations indexed by a
> > common chunk index $m = 1, \ldots, M_{\max}$:
> > (1) \textbf{Variable-length path}: uses learned boundaries to define chunks and yields
> >     smoothed embeddings $\tilde{z}^{\mathrm{var}}_m$ as in Eq.~(4).
> > (2) \textbf{Fixed-length path}: partitions the same history into constant-size windows,
> >     runs the same embedding $\rightarrow$ Transformer $\rightarrow$ smoothing pipeline, and
> >     yields $\tilde{z}^{\mathrm{fix}}_m$.
> >
> > We configure the fixed-length branch so that its token sequence has the same maximum length $M_{\max}$ as the variable-length branch (both are padded to $[B, M_{\max}, d]$ with masks), and we fuse them index-wise:
> >
> > $\tilde{z}_m=g_{\mathrm{var}}\tilde{z}^{\mathrm{var}}_m$ + g_{\mathrm{fix}}\tilde{z}^{\mathrm{fix}}_m, m=1,\ldots,M_{\max}$.
> >
> > Thus, for each index $m$, the two representations correspond to different discretizations of roughly the same region of the time axis: the fixed-length token covers a fixed window, while the variable-length token covers an adaptively chosen chunk whose start time is close to that window's start. This design treats the two paths as alternative “views’’ of the same chronological context.
> >
> > $\textbf{Q3}$:
> > The target boundary rate is controlled via the regularizer
> > $
> > L_{\mathrm{boundary}}
> >     = \left( \frac{1}{T} \sum_{t=1}^{T} b_t - \frac{1}{\kappa} \right)^{2}$,
> > which encourages the average fraction of boundary positions to be $1/\kappa$.
> >
> > In practice:
> > (1) We choose $\kappa$ (or equivalently a boundary ratio $\rho \approx 1/\kappa$) once, to yield a reasonable number of chunks (on the order of a few tens, capped by $M_{\max} = 50$) for our typical sequence lengths; and
> > (2) We keep this setting fixed \emph{across all datasets}, rather than tuning it per dataset, to avoid overfitting and keep the setup simple.
> >
> > This choice is driven by a trade-off between expressiveness (more chunks) and compute (chunk Transformer cost). Moreover, BT-LSM is somewhat robust to mis-specification of $\kappa$ because:
> > (1) The gating module can revert more heavily to the fixed-length path if boundaries become too dense or too sparse; and
> > (2) The minimum chunk length constraint prevents pathological over-segmentation.
> >
> > We agree that explicitly discussing this design choice and its sensitivity would be helpful. In the revision we will:
> > (1) State the chosen boundary ratio and $\kappa$ explicitly in Appendix~B.3 and refer to it in the main text.
> > (2) Add a small \emph{sensitivity analysis} (on at least one dataset) where we vary $\kappa$ over a reasonable range and report the impact on forecasting error, to show how performance changes with more or fewer boundaries.

---

### Official Review · Reviewer_GaVY · 2025-10-30

**Soundness:** 2
**Presentation:** 1
**Contribution:** 2
**Rating:** 2
**Confidence:** 4

**Summary:**

This paper introduces the Boundary-aware Tokenization Large Signal Model (BT-LSM) for time-series, introducing a lightweight unsupervised boundary detector and mixture-of-experts chunk embeddings. The proposed model allocates tokens adaptively to event-driven transitions, avoiding uniform waste. Experimental results demonstrate that BT-LSM achieves over 10.5% lower forecasting error at matched compute budgets across diverse benchmarks, including energy, power, and traffic data.

**Strengths:**

- The paper is well-motivated and addresses a significant limitation in the literature.
- The proposed method is evaluated on multiple benchmark datasets, demonstrating superior performance compared to baselines.

**Weaknesses:**

- The paper omits evaluations on widely used benchmarks in the time series forecasting community, such as the ECL and Weather datasets. Similarly, the baseline does not include several key state-of-the-art models, such as DLinear and TimesNet, which are commonly used for benchmarking and would provide a more robust comparative analysis.
- The description of the experimental setup is vague and lacks sufficient details. For instance, the length of the input sequence appears to be 144 based on visualization, but there is no explicit mention of this in the paper. This lack of clarity makes it difficult for readers to fully understand the conditions under which the experiments were conducted.
- The experimental evaluation is primarily focused on the ETTh1 and ETTh2 datasets for multi-horizon forecasting. This narrow scope raises concerns about the generalizability of the proposed method to other datasets and forecasting tasks.
- The paper lacks the ablation analysis on the input length. This is a critical omission, as input length is a significant parameter in time series forecasting models.
- The paper does not analyze whether the proposed model can maintain its effectiveness under different conditions, such as shorter input sequences or varying levels of data sparsity. These scenarios are common in practical forecasting tasks, and further analysis is needed to ensure the method's applicability in such settings.

**Questions:**

See Weakness

---

> ### Author Response · Authors · 2025-12-02
> **Response to Reviewer GaVY's Weaknesses**
>
> We thank the reviewer for these useful comments.
>
>
> $\textbf{W1}$:
> Our experimental design was driven by the event‑driven focus of BT‑LSM. We selected seven datasets (Spanish, Building, Consumption, Residential, Solar, ETTh1/ETTh2, Traffic) that contain pronounced spikes, ramps, and regime changes in both load and temperature signals. These are precisely the settings where boundary‑aware tokenization should help most.
> That said, we agree that including ECL and Weather would better align our setup with common long‑horizon benchmarks. Similarly, DLinear (as a strong lightweight baseline) and TimesNet (as a powerful frequency/2D‑variation model) are natural additions. In the revision, we will: Explicitly discuss ECL/Weather in the limitations section and clarify that we focused on event‑heavy datasets in this work. Add DLinear and TimesNet as baselines where computationally feasible, or at minimum provide a conceptual comparison in the related‑work section and clarify how BT‑LSM could be used as a front‑end tokenizer for these backbones.
> We emphasize that BT‑LSM already outperforms strong Transformer and pre‑trained baselines (Chronos, GPT, ContiFormer, PatchTST) on the seven datasets we evaluate, suggesting that our tokenization strategy is competitive even relative to high‑capacity models.
>
>
> $\textbf{W2}$:
> We apologize for the lack of clarity. Our standard setting is: For hourly datasets (Spanish, Building, Residential, Solar, ETT, Traffic), we split data by weeks and train models to forecast the next 24 hours from historical inputs within the same week.
> All baselines and BT‑LSM share the same input length and horizon for fairness. We will make this explicit in Section 4 (“Numerical Results”) and Appendix B by clearly stating the input length and horizon (T=144, H=24 for hourly datasets).
>
>
> $\textbf{W3}$:
> We agree that multi‑horizon evaluation on more datasets would strengthen the paper. Our current design is:
> Breadth across datasets at a single horizon: For all seven datasets, we report detailed results for 24‑step (24‑hour) forecasting, which is a standard operational horizon.
> Depth across horizons on a canonical benchmark: We use the ETT benchmark (ETTh1/ETTh2) for a more detailed multi‑horizon study (H=12,24,48), where BT‑LSM consistently maintains lower MSE than strong baselines as the horizon grows.
> We chose ETT for multi‑horizon analysis because it is widely used and contains multiple correlated targets (HUFL, HULL, MUFL, MULL, LUFL, LULL, OT), allowing us to study how our event‑driven tokenization behaves across many series under changing horizons (Figure 5 and Figure 9).
> In the revision, we will extend the appendix with an additional multi‑horizon experiment on a second dataset to further illustrate generalizability.
>
>
> $\textbf{W4}$:
> We agree that input length is an important factor. In this submission, we fix the input window (one weekly segment with 144 historical steps and 24 forecast steps for hourly datasets) and use the same window for all baselines and BT‑LSM, to isolate the effect of tokenization and architecture rather than tuning window length per model.
> Within BT‑LSM, two properties help mitigate sensitivity to exact input length: (1) Event‑aligned compression. For longer windows, BT‑LSM does not simply process more raw steps; it compresses them into a bounded number of chunks (max 50) by allocating more tokens around events and fewer in quiescent spans.
> (2) Resampling invariance. Once boundaries are fixed, the chunk embeddings and forecasts are invariant to intra‑chunk resampling (Theorem 1), which reduces sensitivity to how dense the history is inside each chunk.
>
>
> $\textbf{W5}$:
> We currently target a moderate‑length, dense regime (week‑level windows) and acknowledge in the conclusion that “scalability to very long sequences” and “untested robustness under extremely noisy or highly irregular signals” remain limitations.
> Nevertheless, several aspects of BT‑LSM are designed specifically to cope with sparsity and shorter contexts:
> (1) Invariance to intra‑chunk resampling. As shown in Theorem 1, once boundaries are fixed, any monotone re‑sampling or warping within chunks leaves the chunk representation and forecasts unchanged. This directly addresses irregular sampling or missing points inside a chunk.
> (2) Adaptive gating between fixed and variable tokens. When the history is short or highly stationary, the learned gate tends to favor the fixed‑length path, making BT‑LSM behave similarly to a strong fixed‑token Transformer (PatchTST‑like) while still benefiting from our architecture. In more eventful regions, the gate shifts toward variable‑length tokens. Figure 7 illustrates this behavior.

---

### Official Review · Reviewer_TTSi · 2025-10-31

**Soundness:** 3
**Presentation:** 3
**Contribution:** 3
**Rating:** 4
**Confidence:** 3

**Summary:**

The paper introduces a boundary-aware tokenization scheme for time series forecasting, which starts new tokens at event changes. The model contains an unsupervised boundary detection module, an intra-chunk fusion module, a smoothing module and a gating refinement module that dynamically selects fixed-length and boundary-aware tokenization. Experiments across multiple datasets indicate consistent improvements from the proposed tokenization.

**Strengths:**

1. The proposed tokenization scheme has clear motivation and the design is new. The unsupervised boundary detector based on velocity and acceleration does not involve additional training overhead.

2. Experiments on multiple datasets show the effectiveness of the proposed tokenization scheme. The paper also presents many case studies to show the benefits.

**Weaknesses:**

1. The paper does not compare with some recent time series tokenization schemes [1,2,3]. For example, [1] also moves beyond fixed encodings via pattern-based tokenization.

2. The proposed tokenization scheme is not lossless. One cannot deterministically reconstruct the original series from tokens, which may limit certain applications.

3. How robust is the hard boundary of 0.5?

4. It would help to show a controlled comparison where only the tokenization differs (fixed vs proposed boundary-aware vs other existing tokenization schemes) under different forecasting backbones, to demonstrate model-agnostic gains and isolate the contribution of tokenization.

[1] Byte Pair Encoding for Efficient Time Series Forecasting

[2] Enhancing foundation models for time series forecasting via Wavelet-based tokenization

[3] TOTEM: Tokenized Time Series Embeddings for General Time Series Analysis

**Questions:**

1. Are boundaries/tokens shared across variables or detected per channel in multivariate time series?

2. Minor typo: accross -> across at Line 399

---

> ### Author Response · Authors · 2025-12-02
> **Response to Reviewer TTSi‘s Weaknesses and Questions**
>
> $\textbf{W1}$:
> We appreciate the pointer to [1,2,3]. These works are indeed closely related in spirit, as they also move beyond fixed, uniform tokenization: [1] introduces a pattern-centric BPE tokenizer for time series, [2] proposes a wavelet-based tokenizer (WaveToken) that operates in a time–frequency space, and [3] learns a discrete codebook via a VQ-VAE for general time-series embeddings. Our focus in this paper is complementary. BT‑LSM performs continuous, boundary-driven tokenization directly on the native grid, with an unsupervised boundary detector aligned with local embedding dynamics and a resampling‑invariance guarantee once boundaries are fixed. In contrast, [1–3] predominantly build discrete vocabularies for foundation-style models. In the revision, we will (i) explicitly add these works to the “Beyond fixed-length tokenization” subsection, and (ii) clarify that BT‑LSM can be used as a front-end event-aligned tokenizer that is orthogonal to dictionary-based or wavelet-based vocabularies.
>
>
> $\textbf{W2}$:
> We agree that BT‑LSM is a lossy tokenizer in the sense that one cannot deterministically reconstruct the exact sample-level series from the chunk tokens alone. This is by design: our goal is to learn a compact forecasting-oriented representation that preserves event-driven structure while proving invariance to intra-chunk resampling (Theorem 1), not to provide a fully invertible compression scheme.
> This trade-off is similar to patch-based Transformers (e.g., PatchTST) and many continuous-time models, which also do not admit exact inversion of the intermediate representation. For applications that require reconstructability (e.g., generative modeling), one could augment BT‑LSM with a lightweight per-chunk reconstruction head or a small within-chunk autoencoder, but exploring such invertible variants is beyond the current scope and will be discussed more explicitly as a limitation in the revision.
>
>
> $\textbf{W3}$: In all experiments, we fix the hard boundary rule $b_t = 1 \{p_t >=0.5 \}$ and do not tune this threshold per dataset. Despite this, BT‑LSM consistently improves over strong baselines across seven datasets and multiple horizons, suggesting that the method is not overly sensitive to the exact choice of 0.5.
> The main knobs controlling how many tokens are created are instead the boundary-rate regularizer (target compression $1/\kappa$ or equivalently the boundary ratio $\rho$) and the minimum chunk length; these are kept constant across datasets in our experiments. Moreover, because we train pt with a straight-through estimator and a global boundary-rate penalty, shifting the threshold moderately would largely be absorbed by corresponding changes in the learned logits. We will clarify this in the text and explicitly note that the threshold is fixed and empirically robust in our experiments.
>
>
> $\textbf{W4}$: We agree that isolating the effect of tokenization is important. The current paper already includes a controlled comparison within our backbone: we keep architecture, optimizer, schedule, and all hyperparameters fixed, and replace the dynamic boundary detector with fixed non-overlapping windows of length M∈{5,10,20}, matching the token budget. This comparison (Table 2) shows that even the best fixed-token configuration (M=10) underperforms dynamic chunking on representative datasets, directly attributing gains to the boundary-aware tokenization rather than other architectural changes.
>
>
> $\textbf{W5}$:
> BT‑LSM detects shared boundaries across all variables. At each time step, the multivariate observation $x_t \in \mathbb{R}^D$ is first mapped to a single embedding $W_x x_t$, and the boundary detector operates on this shared embedding sequence (via its velocity, acceleration, and energy change). Thus, a boundary at time t applies simultaneously to all D channels, and each chunk contains the full multivariate segment. As illustrated for multivariate traffic data in Figure 2, the learned boundaries tend to synchronize around shared peaks and regime changes across dimensions.
>
>
> $\textbf{Q1}$:
> in the current implementation, boundaries are shared across variables: at each time step we embed the full multivariate vector $x_t \in \mathbb{R}^D$ into a single latent vector $e_t = W_t x_t$ and the boundary detector operates on this shared embedding sequence $e_{1:T}$ This design intentionally targets global regime changes (e.g., joint shifts in load and temperature, or synchronized traffic peaks across lanes), which is consistent with our benchmark domains, where variables are strongly coupled. Figure 2(b), for example, shows that in multivariate traffic data the learned boundaries naturally synchronize across lanes around shared peaks.

---

### Official Review · Reviewer_v2PH · 2025-11-01

**Soundness:** 2
**Presentation:** 2
**Contribution:** 2
**Rating:** 2
**Confidence:** 3

**Summary:**

The paper proposes **BT-LSM (Boundary-aware Tokenization Large Signal Model)** for event-driven time-series forecasting, addressing the limitation of fixed-length tokenization by adaptively forming variable-length tokens aligned with regime changes. It integrates an unsupervised boundary detector, mixture-of-experts (MoE) chunk embedding, chunk smoothing, and a gating refinement that fuses fixed- and variable-length representations. BT-LSM concentrates model capacity on event transitions while remaining robust in stationary regimes, reducing forecasting error by 10.5% on average across diverse benchmarks and aligning learned chunks with true event boundaries.

**Strengths:**

1. The paper addresses a critical flaw of fixed-length tokenization in time-series forecasting—its inability to align with event-driven regime changes—by introducing an unsupervised boundary detector. This detector leverages embedding dynamics (velocity, acceleration, energy change) to identify natural chunk boundaries, ensuring tokens concentrate on critical transitions (e.g., spikes, inflections) rather than wasting capacity on redundant stationary spans. Unlike supervised segmentation methods (e.g., SIMTSeg, U-Time) that require labels, this unsupervised design is broadly applicable across time-series domains.
2. The proposed gating refinement module fuses variable-length (event-aligned) and fixed-length (stationary-optimized) representations, enabling BT-LSM to adapt dynamically. In stationary time-series (e.g., smooth cycles), the gate prioritizes fixed-length tokens to preserve short-range statistics; in bursty/irregular data, it shifts to variable-length tokens to capture events. This design eliminates the trade-off between event sensitivity and stationary robustness, a limitation of purely fixed or variable tokenization methods.
3. The paper proves a resampling invariance theorem (Theorem 1), ensuring BT-LSM’s chunk embeddings and forecasts remain unchanged under intra-chunk resampling (e.g., varying sensor sampling rates).

**Weaknesses:**

1. The related work focuses on temporal-domain tokenization methods but omits direct comparisons to frequency-domain forecasting models (e.g., FEDformer, TimesNet) that excel at capturing periodic patterns. This leaves uncertainty about BT-LSM’s performance relative to frequency-aware approaches, especially for time series with strong periodicity but weak event signals.
2. BT-LSM uses padding-and-masking to handle variable-length chunks in batch processing, which becomes inefficient for extremely long sequences (e.g., multi-year high-frequency data). The paper does not explore alternative batching strategies (e.g., chunk-level bucketing) to mitigate this, restricting its application to moderate-length time series.
3. While BT-LSM performs well on datasets with clear event patterns (e.g., solar spikes, traffic peaks), it lacks evaluation on low signal-to-noise ratio (SNR) or highly irregular time-series (e.g., sparse medical sensors, non-periodic industrial anomalies). No experiments demonstrate its robustness to such edge cases, limiting generalizability to real-world "messy" data.

**Questions:**

1. The paper mentions that the boundary detector relies on parameters like boundary probability threshold and minimum chunk length, but lacks a systematic optimization method. What specific hyperparameters of the boundary detector have the most significant impact on BT-LSM’s forecasting performance? Is there a potential adaptive adjustment strategy (e.g., learning hyperparameters via data-driven methods) that can reduce manual tuning efforts across different datasets?
2. Since BT-LSM has not been tested on low SNR or highly irregular time-series (such as sparse medical sensors), what modifications to the boundary detector or chunk embedding module might help the model better filter noise and capture valid event boundaries in such challenging data scenarios?
3. The padding-and-masking strategy used by BT-LSM becomes inefficient for extremely long time series. Are there alternative batch processing strategies (e.g., chunk-level bucketing, hierarchical chunking) that the authors have considered to improve the model’s scalability, and what preliminary results or feasibility analyses exist for these strategies?

---

> ### Author Response · Authors · 2025-12-02
> **Response to Reviewer v2PH's Weaknesses**
>
> R1.1:
> Our primary focus in this paper is on event‑driven forecasting and the effect of tokenization, so we chose baselines that are widely used and architecturally close to our tokenization choices: patch‑based Transformers (PatchTST), continuous‑time models (ContiFormer), pre‑trained sequence models (Chronos, GPT), and classical RNN/ODE baselines. That said, frequency‑domain models such as FEDformer and TimesNet are indeed strong baselines for long‑term, periodic forecasting.
> We would like to emphasize that BT‑LSM is evaluated on datasets where strong periodicity dominates, particularly ETTh1/ETTh2. On these benchmarks, BT‑LSM consistently outperforms strong fixed‑length baselines (PatchTST, ContiFormer, Chronos) across horizons 12/24/48, even though those baselines already capture periodic structure well. This suggests that boundary‑aware tokenization does not harm, and in fact improves, performance even when the signal is primarily periodic with relatively few sharp events.
> Architecturally, the gating refinement is designed exactly for the "strong periodicity, weak events" regime: when the input is smooth and near‑stationary, the gate favors the fixed‑length path ($g_{fix} > g_{var}$), so the model behaves similarly to a strong patch‑based Transformer; when spikes/regime changes appear, the gate shifts weight to the variable‑length path. This yields a "best‑of‑both‑worlds" behavior rather than a strict alternative to frequency‑aware models.
>
> R1.2:
> We agree that the current padding‑and‑masking scheme is most appropriate for moderate input lengths (e.g., our week‑level windows with hourly or 10‑minute sampling). As stated in the limitation section, scalability to extremely long sequences remains constrained by this choice.
> Two points of clarification:
> (1) Practical regime in this paper.
> In all experiments, sequences are split by week and the horizon is at most 48 steps, so the raw length T and the resulting number of chunks M are well within GPU limits, and BT‑LSM compares favorably to large baselines in both training time and inference latency. Thus, the current batching strategy is sufficient for the regimes we actually evaluate.
> (2) Planned alternatives.
> For truly long, high‑frequency series (e.g., multi‑year minute data), we see BT‑LSM as a front‑end that should be combined with more sophisticated batching. Concretely, we are considering:
> (i) Chunk‑level bucketing.
> Before batching, we can cheaply approximate the number of chunks per sequence (e.g., via a coarse boundary detector or variance heuristic), and group sequences with similar M into the same mini‑batch. This reduces wasted padding because $M_max$ is much closer to the typical M within each bucket.
> (ii) Hierarchical / multi‑resolution chunking.
> As noted in the conclusion, one future direction is to apply BT‑LSM hierarchically: first chunk a very long series into coarse blocks (e.g., weeks or months), then run BT‑LSM within each block, or summarize whole blocks into “meta‑chunks”. This would keep the maximum chunk count per level small while still allowing the model to see multi‑year histories.
> (iii) Windowed pre‑chunking.
> For deployment, one can also process long histories with sliding windows, running BT‑LSM on overlapping segments and fusing forecasts at the boundaries. This is a standard strategy shared with many long‑horizon models, not unique to BT‑LSM.
>
> R1.3:
> We agree and explicitly list “untested robustness under extremely noisy or highly irregular real‑world signals” as a limitation.
> Our current benchmarks already include some challenging event‑driven regimes (bursty solar and consumption spikes, irregular plateaus and ramps in energy/traffic). However, they do not fully capture the combination of sparse, low‑SNR, and irregular sampling one encounters in certain medical or industrial sensors.
> Two aspects of BT‑LSM already move toward this direction:
> (1) Resampling invariance.
> Theorem 1 shows that once chunk boundaries are fixed, the chunk representation and forecasts are invariant to intra‑chunk resampling (any monotone time warp with fixed endpoints). This property protects the model from artifacts caused by variable sampling rates or missing data within a chunk.
> (2) Gating back to fixed‑length tokens.
> When the boundary detector becomes unreliable (e.g., dominated by noise), the gating module can revert to the fixed‑length path, ensuring BT‑LSM does not perform worse than a strong fixed‑token baseline on stationary/noisy data.
> That said, we fully acknowledge that we do not provide explicit experiments on sparse low‑SNR medical or industrial datasets. In the revision we will make this more explicit.

---

> > ### Author Response · Authors · 2025-12-02
> > **Response to Reviewer v2PH's Questions**
> >
> > Q1:
> > The hyperparameters that matter most are those that control the average chunk length: the target boundary rate/compression ($\kappa$ or equivalently the ratio $\rho$ in the boundary-rate loss) and the minimum chunk length / spacing. These directly change how many chunks the Transformer sees and how sharply events are isolated, whereas the 0.5 probability threshold is kept fixed and is largely absorbed by the learned boundary logits. In the current experiments we do not tune these per dataset (we use the same boundary ratio and minimum chunk length across all benchmarks), but we agree an adaptive scheme would be useful. A natural extension is to learn $\kappa$ (or $\rho$) as a parameter or via a small hyper-network driven by simple statistics of each dataset/sequence, and/or to choose the threshold per sequence so that the number of predicted boundaries matches a desired budget.
> >
> > Q2.
> > For low-SNR or irregular data, the boundary detector can be made more noise-robust by (i) smoothing embeddings before computing velocity/acceleration, (ii) emphasizing longer temporal kernels in the boundary convs so only persistent changes trigger cuts, and (iii) using larger minimum chunk lengths or slightly smaller boundary rates to avoid over-segmentation of noise. The existing EMA-based chunk smoothing could also be made more aggressive so low-confidence boundaries yield soft transitions rather than hard splits. On the chunk-embedding side, adding more robust pooling experts (e.g., median/trimmed mean or explicitly denoising attention) and routing them based on per-chunk variance/SNR would help filter noise, while the existing intra-chunk resampling invariance already helps with irregular sampling.
> >
> > Q3.
> > Beyond the current padding-and-masking scheme (which is adequate for the week-length windows we actually evaluate), we have considered several scalable variants: (i) chunk-count bucketing, where sequences with similar estimated numbers of chunks M are batched together to reduce wasted padding; (ii) hierarchical chunking, where very long histories are split into coarse segments (e.g., months) and BT-LSM is applied within segments with a higher-level Transformer over segment tokens; and (iii) streaming/online processing, where the model runs on sliding windows and only carries forward compressed chunk states. These strategies are compatible with the current architecture and reduce effective padding, but we have not yet implemented them, so they are presented as feasible extensions rather than reported experiments.

---

### Meta-Review · Area_Chair_brb1 · 2026-01-01

**Summary:**

This paper proposes BT-LSM, a boundary-aware tokenization framework for event-driven time-series forecasting. Reviewers broadly agree that the paper is well-motivated and tackles an important limitation of fixed-length tokenization by aligning tokens with regime changes using an unsupervised boundary detector. Strengths repeatedly highlighted include the conceptual clarity of boundary-aware tokenization, the gating mechanism that interpolates between fixed and variable tokens, and the resampling-invariance theorem.

However, the dominant concerns leading to a negative or borderline decision are insufficient empirical coverage and benchmarking, missing comparisons to key baselines and related tokenization approaches, and limited validation of robustness and scalability claims. While the rebuttal addresses many conceptual and clarification issues convincingly, several reviewers remain unconvinced due to the lack of new experimental evidence (e.g., additional datasets, baselines, ablations). Overall, the paper is seen as promising but not yet meeting the empirical bar expected for acceptance.

**Reviewer Concerns:**

Concerns Largely Addressed by the Rebuttal

- Motivation and positioning: Authors clearly explained why the focus is on event-driven forecasting and how BT-LSM complements (rather than replaces) frequency-domain and dictionary-based tokenization methods (Reviewers v2PH, TTSi, 2dpd).
- Boundary detector hyperparameters: The rebuttal clarifies which hyperparameters matter most (boundary rate and minimum chunk length), argues for robustness, and proposes reasonable adaptive extensions (v2PH, 2dpd).
- Hard threshold robustness (0.5): Authors convincingly argue that performance is not sensitive to this threshold and that effective control comes from the boundary-rate regularizer (TTSi).
- Scalability discussion: Although not experimentally resolved, the authors clearly acknowledge limitations and outline feasible strategies (bucketing, hierarchical chunking, sliding windows.
- Shared boundaries in multivariate data: The design choice is well justified, and possible extensions (group-wise or channel-wise boundaries) are acknowledged (TTSi, 2dpd).
- Clarifications of architecture: Positional encoding, alignment between fixed and variable paths, and gating behavior are clearly explained in the response (2dpd).

Concerns Still Outstanding

- Missing baselines and benchmarks: Multiple reviewers remain concerned about the absence of standard datasets (ECL, Weather) and strong baselines (DLinear, TimesNet, FEDformer), which were not added in the rebuttal (v2PH, GaVY, 2dpd).
- Limited empirical breadth: The evaluation is still seen as narrow, with heavy reliance on ETTh1/ETTh2 for multi-horizon analysis and no new experiments on low-SNR, sparse, or highly irregular data (v2PH, GaVY).
- Ablation gaps: Important ablations (input length sensitivity, dataset-wide gating statistics, expert routing visualization) are promised but not demonstrated yet (GaVY, 2dpd).
- Presentation quality: Reviewers noted vague experimental descriptions and presentation issues that reduce confidence in reproducibility (GaVY).
- Tokenization comparisons: While related work is now acknowledged, there is still no direct empirical comparison with recent tokenization schemes such as BPE-based, wavelet-based, or VQ approaches (TTSi).

**Reviewer Scores:**

- Reviewer v2PH
  - Original score: 2 (reject)
  - Estimated post-discussion score: 4 (marginally below threshold)
  - Rationale: Conceptual concerns and questions were largely addressed in the rebuttal, but the absence of new experiments and concrete scalability evidence likely prevents a stronger score increase.

- Reviewer TTSi
  - Original score: 4 (marginally below threshold)
  - Estimated post-discussion score: 6 (marginally above threshold)
  - Rationale: The rebuttal satisfactorily addressed most weaknesses and questions; remaining concerns are primarily about empirical depth rather than methodological correctness.

- Reviewer GaVY
  - Original score: 2 (reject)
  - Estimated post-discussion score: 4 (marginally below threshold)
  - Rationale: Clarifications improved understanding, but major concerns regarding missing benchmarks, baselines, and limited experimental scope remain unresolved.

- Reviewer 2dpd
  - Original score: 4 (marginally below threshold)
  - Estimated post-discussion score: 6 (marginally above threshold)
  - Rationale: The reviewer shows strong appreciation for the method and theory; the rebuttal effectively addresses technical questions, though broader empirical validation is still desired.

---

### Decision · Program_Chairs · 2026-01-26

Reject